# Striking parallels between dorsoventral patterning in *Drosophila* and *Gryllus* reveal a complex evolutionary history behind a model gene regulatory network

Matthias Pechmann[1]*, Nathan James Kenny[2†], Laura Pott[1], Peter Heger[3], Yen-Ta Chen[1], Thomas Buchta[1], Orhan Özüak[1], Jeremy Lynch[1,4], Siegfried Roth[1]*

[1]Institute for Zoology/Developmental Biology, Biocenter, University of Cologne, Köln, Germany; [2]The Natural History Museum, London, United Kingdom; [3]Regional Computing Centre (RRZK), University of Cologne, Köln, Germany; [4]Department of Biological Sciences, University of Illinois at Chicago, Chicago, United States

*For correspondence:
pechmanm@uni-koeln.de (MP);
siegfried.roth@uni-koeln.de (SR)

Present address: [†]
Departmentof Biological and
Medical Sciences, Oxford
Brookes University, Oxford,
United Kingdom

Competing interests: The
authors declare that no
competing interests exist.

Reviewing editor: Patricia J
Wittkopp, University of
Michigan, United States

**Abstract** Dorsoventral pattering relies on Toll and BMP signalling in all insects studied so far, with variations in the relative contributions of both pathways. *Drosophila* and the beetle *Tribolium* share extensive dependence on Toll, while representatives of more distantly related lineages like the wasp *Nasonia* and bug *Oncopeltus* rely more strongly on BMP signalling. Here, we show that in the cricket *Gryllus bimaculatus*, an evolutionarily distant outgroup, Toll has, like in *Drosophila*, a direct patterning role for the ventral half of the embryo. In addition, Toll polarises BMP signalling, although this does not involve the conserved BMP inhibitor Sog/Chordin. Finally, Toll activation relies on ovarian patterning mechanisms with striking similarity to *Drosophila*. Our data suggest two surprising hypotheses: (1) that Toll's patterning function in *Gryllus* and *Drosophila* is the result of convergent evolution or (2) a *Drosophila*-like system arose early in insect evolution and was extensively altered in multiple independent lineages.

## Introduction

In all insects studied so far, Toll and BMP signalling are essential for establishing the dorsoventral (DV) body axis of the embryo (*Lynch and Roth, 2011*; *Özüak et al., 2014b*; *Sachs et al., 2015*). Toll signalling acts ventrally and is involved in specifying the mesoderm and neurogenic (ventral) ectoderm. BMP signalling acts dorsally and is required for specifying the extraembryonic tissues and the non-neurogenic (dorsal) ectoderm. Although the relationship between pathway activity and DV cell fates is largely conserved, the interplay between both pathways and the amount of spatial information provided by each pathway have changed dramatically during insect evolution (*Figure 1*).

In *Drosophila*, Toll signalling tightly controls all aspects of DV axis formation. Maternally provided eggshell cues, which depend on the ventral expression of *pipe* in the ovarian cells producing the eggshell (so-called follicle cells), determine the shape of a steep, long-range Toll signalling gradient (*Figure 1*). This gradient acts as a morphogen that provides precise spatial information along the entire DV axis (*Moussian and Roth, 2005*; *Schloop et al., 2020*; *Stein and Stevens, 2014*). Toll signalling regulates the expression of more than 50 target genes in a concentration-dependent manner (*Hong et al., 2008*; *Reeves and Stathopoulos, 2009*).

On the one hand, Toll has a direct instructive role in specifying the mesoderm and ventral parts of the neuroectoderm. It activates genes that specify these fates, and these Toll target genes are not affected by loss of BMP signalling (*Mizutani et al., 2006*; *von Ohlen and Doe, 2000*; *Figure 1*). On the other hand, Toll tightly controls BMP signalling and directly regulates at least five

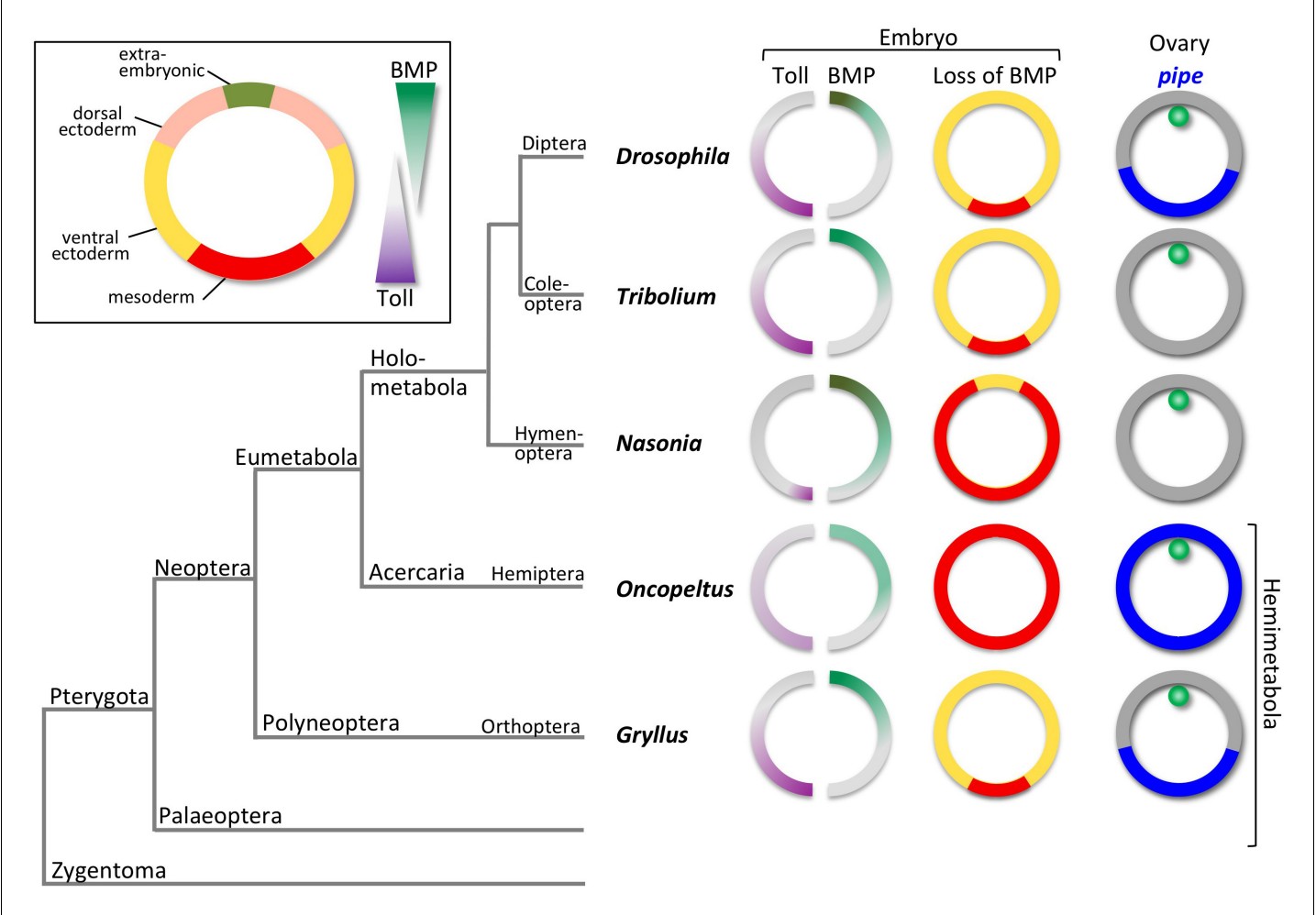

**Figure 1.** The evolution of dorsoventral (DV) patterning in insects. The upper-left frame shows a cross section through a blastoderm embryo, indicating the prospective regions along the DV axis. The gradient filled triangles indicate the two major signalling pathways required for establishing these fates in most insect embryos. The phylogenetic tree shows the relationships of five insects for which the DV patterning system has been analysed at the functional level, including *Gryllus*, the subject of this paper. For each insect, key findings are depicted schematically using cross sections through blastoderm embryos and egg chambers. The sections on the left show half-embryos with Toll (left) and BMP (right) signalling gradients. The depicted distribution of Toll signalling is based on indirect evidence in the case of *Nasonia*, *Oncopeltus* and *Gryllus*. The middle column of sections shows the fate map shift upon loss of BMP signalling uncovering the BMP-independent patterning functions of Toll signalling. The right column shows the follicular epithelium surrounding the oocyte with the oocyte nucleus in green. *pipe* expression is indicated in blue.

components of the BMP regulatory network. Crucially these include the major BMP ligand (*decapentaplegic*, *dpp*) and the BMP inhibitor *short gastrulation* (*sog*), which is the homolog of vertebrate *chordin* (**Bier and De Robertis, 2015**; **Jaźwińska et al., 1999**; **O'Connor et al., 2006**). These Toll-dependent regulatory inputs initiate the formation of a BMP gradient which is in turn required to specify dorsal (amnioserosa) and dorsolateral (non-neurogenic ectoderm) cell fates and to restrict ventrolateral cell fates (**Mizutani et al., 2006**; **O'Connor et al., 2006**).

Comprehensive functional studies on DV axis formation have so far been performed for three other insect species. Two of these (the flour beetle *Tribolium castaneum* and the parasitoid wasp *Nasonia vitripennis*) are holometabolous insects, meaning that, like *Drosophila*, they undergo complete metamorphosis. The other species (the milkweed bug *Oncopeltus fasciatus*) is hemimetabolous and undergoes incomplete metamorphosis.

*Tribolium* uses the same basic regulatory logic for setting up the embryonic DV axis as *Drosophila*: the mesodermal primordium is specified by Toll signalling alone and is not affected by a loss of BMP activity (**Nunes da Fonseca et al., 2010**; **van der Zee et al., 2006**; **Figure 1**). In addition to its

ventral patterning role, Toll polarises the BMP gradient by activating the BMP inhibitor *sog*. A main difference to *Drosophila* is the lack of extensive maternal pre-patterning. Toll is expressed zygotically, and its activation does not occur via *pipe*-dependent eggshell components (*Chen et al., 2000*; *Lynch et al., 2010*). Indirect evidence suggests that patterning starts with weakly asymmetric spatial cues. Subsequently, a combination of positive and negative feedback of Toll pathway components leads to the formation of a dynamically changing long-range Toll signalling gradient (*Nunes da Fonseca et al., 2008*).

In the wasp *Nasonia*, Toll is also required to initiate mesoderm formation. However, the size of the mesodermal domain depends on repression by BMP (*Özüak et al., 2014b*). Thus, in *Nasonia*, BMP not only controls dorsal and lateral, but also ventral cell fates. Moreover, BMP gradient formation does not depend on Toll signalling, but has its own dorsally localised, maternal source (*Özüak et al., 2014a*). Ventral Toll activation in *Nasonia* occurs with high spatial precision through unknown molecular mechanisms (*Figure 1*; *Özüak et al., 2014b*).

Finally, in *Oncopeltus*, Toll signalling lacks direct patterning functions altogether. It is not even strictly required for mesoderm formation, which can be achieved just by suppression of BMP activity (*Sachs et al., 2015*). Toll signalling appears to be weakly polarised by eggshell components requiring follicle cell expression of *pipe*. However, *Oncopeltus pipe* lacks detectable DV asymmetry (*Figure 1*; *Chen, 2015*). Polarised Toll signalling enhances ventral *sog* expression which is part of a self-regulatory BMP signalling network that patterns the entire DV axis (*Sachs et al., 2015*).

Given the phylogenetic relationship of the species described above (*Figure 1*), the most parsimonious explanation for the evolution of DV patterning is that the strong dependence on Toll arose in a common ancestor of *Tribolium* and *Drosophila*, while the ancestral state, maintained in *Nasonia* and *Oncopeltus*, was characterised by a larger dependence on BMP signalling. Toll has not been found to be involved in DV axis formation outside the insects, while BMP signalling is essential for DV patterning in most metazoans including other arthropod classes (*Akiyama-Oda and Oda, 2006*; *De Robertis, 2008*; *Saina et al., 2009*). Therefore, it is possible that the recruitment of Toll signalling for axis formation occurred within or at the base of the insect stem lineage. Our observed trend might thus reflect steps of the progressive recruitment of Toll for axis formation. However, this evolutionary scenario is based on a very limited number of species with large parts of the insect evolutionary tree still unexplored (*Figure 1*). In particular, representatives of the other prominent branch of the Neoptera (the Polyneoptera), and of the Palaeoptera, are crucial for reconstructing the ancestral mechanism and evolutionary history of the insect DV patterning system.

To partially fill this gap, we have analysed DV patterning in the polyneopteran species *Gryllus bimaculatus*. Based on new transcriptome data, we have searched for major components of the Toll and BMP signalling pathways. Our functional analyses with these components show, in contrast to *Nasonia* or *Oncopeltus*, that BMP signalling in *Gryllus* is not required to restrict ventralmost cell fates (*Figure 1*). BMP's function appears to be confined to the dorsal half of the embryo, while patterning of the ventral half relies on Toll signalling alone, like in *Drosophila* and *Tribolium*. Toll signalling is also required to polarise BMP signalling, which constitutes another similarity to *Drosophila* and *Tribolium*. In addition, *Gryllus* Toll signalling is polarised by eggshell cues that depend on localised *pipe* expression in the follicular epithelium, a mechanism so far only observed in *Drosophila* (*Figure 1*), further highlighting the profound similarities between *Gryllus* and *Drosophila*. However, we do detect significant divergences as well, such as the loss of the critical BMP inhibitor *sog/chordin* in *Gryllus*. Overall our analyses point to the possibility that the striking and surprising similarities between fly and cricket are due to convergent evolution of several developmental characters rather than conservative maintenance of an ancestral state lost in several other insect lineages.

## Results

### A novel reference transcriptome to analyse ovary and embryonic development in *G. bimaculatus*

To better analyse embryonic development and search for components that could be required to pattern the DV body axis of *G. bimaculatus,* we sequenced the transcriptomes of ovaries and embryonic egg stages 1–12 (staging according to *Donoughe and Extavour, 2016*). Combining our sequencing

reads with previous sequencing reads from transcriptomic studies of *G. bimaculatus* (*Bando et al., 2013*; *Fisher et al., 2018*; *Zeng et al., 2013*), we generated a novel reference transcriptome.

Statistics related to our cleaned reference assembly can be seen in *Table 1*. A total of 328,616 contigs are present, and Trinity, a splice aware assembler, automatically assigned these to 244,946 independent 'gene' clusters. The contigs are well-assembled, with a N50 (2134 bp) more than sufficient to represent the full length of most protein coding genes. 34,565 contigs were present at equal or greater length than this N50 value, and 69,081 were longer than 1 kb in length. Our assembly results are similar in some metrics to those obtained by *Fisher et al., 2018*, although our slightly better N50 and mean length figures, coupled to a lower number of transcripts (328,616 *cf.* 511,724) and a much higher number of bases contained in our assembly (301,016,284 *cf.* 237,416,984), suggest that our assembly is more contiguous.

To assay the completeness of our transcriptomic assembly, we compared our dataset against the BUSCO ('Basic Universal Single Copy Orthologue') set of highly conserved genes using the metazoan and eukaryote complements (978 and 303 genes, respectively). Our results show almost total recovery of these cassettes. Of the eukaryotic BUSCO set of 303 genes, none were missing. 85 were complete and single copy, 215 were duplicated (likely representing isoformal variants in this mixed transcriptome) and 3 were present in fragments. The 978-gene metazoan BUSCO set is almost complete. 966 of these genes are present in full length (665 as duplicates), and 11 are present but fragmented. Only one gene is missing (*Riboflavin kinase*). The low number of fragmented genes strongly suggests that our transcriptome is well assembled, and the low number of missing genes suggests that almost the entirety of the transcribed portion of the genome is captured by our sampling. Our resource was annotated using an automated Blast approach. 101,760 of the 328,616 contigs were annotatable by Diamond BlastX with an *E* value cutoff of $10^{-6}$. The results of these automatic annotations are recorded in *Supplementary file 2*.

## The TGFβ and Toll pathway components of *G. bimaculatus*

We used our reference transcriptome to systematically search for components of the TGFβ and Toll signalling pathways and compared our findings for *Gryllus* in particular with *Drosophila*, *Tribolium*, *Nasonia* and *Oncopeltus*, the insects for which the DV patterning roles of both pathways have been most thoroughly studied (*O'Connor et al., 2006*; *Özüak et al., 2014a*; *Özüak et al., 2014b*; *Panfilio et al., 2019*; *Sachs et al., 2015*; *Van der Zee et al., 2008*; *van der Zee et al., 2006*). The sequences of these genes, and alignments used for phylogenetic reconstruction, can be found in *Supplementary file 3*.

The TGFβ pathway consists of extracellular ligands and their modulators, transmembrane receptors, cytoplasmic signal transducers and downstream transcription factors (*Balemans and Van Hul, 2002*; *Parker et al., 2004*; *Upadhyay et al., 2017*). *Gryllus* has homologs of all TGFβ ligands found in *Drosophila*, *Tribolium*, *Nasonia* and *Oncopeltus* (*Figure 2—figure supplement 1*). *Oncopeltus* and *Nasonia* have several Dpp/BMP2,4 homologs. Likewise, we identified two Dpp/BMP2,4 homologs in *Gryllus*, confirming previous studies (*Donoughe et al., 2014*; *Ishimaru et al., 2016*).

**Table 1.** Summary statistics: assembly.

| Number of Trinity transcripts | 328,616 |
| --- | --- |
| Number of Trinity 'genes' | 244,946 |
| Min contig length | 201 |
| Max contig length | 32,386 |
| Mean contig length | 916.01 |
| N50 contig length | 2134 |
| Number of contigs ≥ 1 kb | 69,081 |
| Number of contigs in N50 | 34,565 |
| Number of bases in all contigs | 301,016,284 |
| Number of bases in contigs ≥ 1 kb | 200,921,630 |
| GC content of contigs (%) | 39.72 |

However, there was only a single Gbb/BMP5-8 homolog, a ligand that is duplicated in *Drosophila*, *Tribolium* and *Nasonia*. Some ligands are only present in particular lineages. Like *Tribolium*, *Gryllus* possesses a homolog of BMP9, a ligand whose function has only been characterised in vertebrates so far. Like *Nasonia*, *Gryllus* possesses a homolog of ADMP, a BMP-like ligand that is involved in size regulation during early DV axis formation in vertebrates (*Leibovich et al., 2018*). Interestingly, unlike any other insect studied so far *Gryllus* possess a homolog of BMP3, a ligand with inhibitory effects on BMP signalling (*Daluiski et al., 2001*; *Figure 2—figure supplement 1*, *Supplementary file 4*).

*Gryllus* has the complete complement of TGFβ receptors and cytoplasmic signal transducers found in other insects, including single homologs of Punt, Thickveins and Saxophone, the SMAD proteins Mad, Medea and Dad, the transcription factors Schnurri and Brinker (*Donoughe et al., 2014*; *Ishimaru et al., 2016*), as well as the E3 ligase SMURF (which downregulates Medea) (*Podos et al., 2001*) and the inhibitory transmembrane protein Pentagone (*Norman et al., 2016*). However, *Gryllus* lacks the inhibitory BMP receptor BAMBI (*Onichtchouk et al., 1999*), which is present in *Tribolium*, *Nasonia* and *Oncopeltus* (*Figure 2—figure supplements 2–5*, *Supplementary file 4*).

The most surprising difference to other insects pertains to the level of extracellular BMP modulators. In particular, we were unable to identify a homolog of Sog/Chordin within either our trancriptome or the *G. bimaculatus* genome (*Ylla et al., 2020*). In addition, we could not find Sog in the transcriptome of *Gryllus rubens* (*Berdan et al., 2016*) and failed to clone *sog* from *G. bimaculatus* and another cricket species (*Acheta domesticus*) via degenerate PCR. This came as a surprise as Sog is an antagonist of the BMP signalling pathway that is conserved from coelenterates to vertebrates (*De Robertis and Moriyama, 2016*; *Saina et al., 2009*), and it is present in the genomes of several polyneopteran insect species, like the stick insect *Timema cristinae* (our own BLAST search in the NCBI BioProject: PRJNA357256), the cockroach *Blattella germanica* (GenBank: PSN34641.1) and the termite *Zootermopsis nevadensis* (NCBI Reference Sequence: XP_021915980.1). We generated a hidden Markov model trained to identify insect Sog protein sequences and used it to search for Sog homologs in available orthopteran transcriptomes (*Figure 2*, *Supplementary file 1*). Transcripts containing *sog*-like sequences were found in representatives of both major branches of Orthoptera, the Caelifera (e.g. *Locusta*) and the Ensifera (e.g. *Distrammena*); however, none of the six representatives of the Gryllidae possessed transcripts with significant Sog homology. This observation suggests that Sog was lost within the Ensifera, in the lineage leading to the Gryllidae.

We also have observed an independent loss of *Sog* in the wasp *N. vitripennis* (*Özüak et al., 2014a*; *Özüak et al., 2014b*), showing that this seemingly crucial factor can be lost surprisingly readily in the course of evolution. However, as discussed later, the DV patterning context of *Nasonia* and *Gryllus* are radically different.

Despite the lack of Sog, *Gryllus* possesses several extracellular proteins that are known to interact with Sog/Chordin in *Drosophila* and vertebrates (*Balemans and Van Hul, 2002*; *Parker et al., 2004*; *Upadhyay et al., 2017*). These include the metalloprotease Tolloid (which normally cleaves Sog), Tsg and CV2, all of which are thought to be present in complexes with BMP and Sog (*Figure 2—figure supplement 6*). However, these components have been shown to also fulfil Sog-independent functions in *Nasonia* (*Özüak et al., 2014a*; *Özüak et al., 2014b*). *Gryllus* also possesses a range of other BMP inhibitors aside from Sog, including Follistatin (also present in *Drosophila*, *Tribolium* and *Nasonia*), Noggin (so far seen only found in *Oncopeltus* and other hemimetabolous insects), Dan and Gremlin (absent in *Oncopeltus*, but present in *Tribolium*) (*Figure 2—figure supplements 5* and *7*, *Supplementary file 4*).

The activation of the Toll pathway during DV patterning in *Drosophila* requires eggshell components that are produced by the direct or indirect action of three genes: *nudel (ndl)*, *pipe (pip)* and *windbeutel (wind)* (*Stein and Stevens, 2014*). Homologs of all three genes were identified in our *Gryllus* transcriptome (*Figure 2—figure supplement 8*, *Supplementary file 5*). In *Drosophila*, these eggshell components trigger the activation of a proteolytic cascade that is required to produce active Spätzle protein, which is the ligand of the Toll receptor (*Stein and Stevens, 2014*). While the members of the proteolytic cascade (*gastrulation defective*, *easter* and *snake*) could not be unequivocally identified, we found two *spätzle* homologs and one homolog of *Toll1*, the Toll receptor involved in DV patterning and innate immunity (*Figure 2—figure supplement 9*, *Supplementary file 5*; *Benton et al., 2016*). A highly conserved set of components acts downstream of Toll: the adaptor protein Myd88, the kinases Tube and Pelle, the cytoplasmic inhibitor Cactus/I-kB and the NF-κB transcription factor Dorsal (*Schloop et al., 2020*). While all the former have a unique

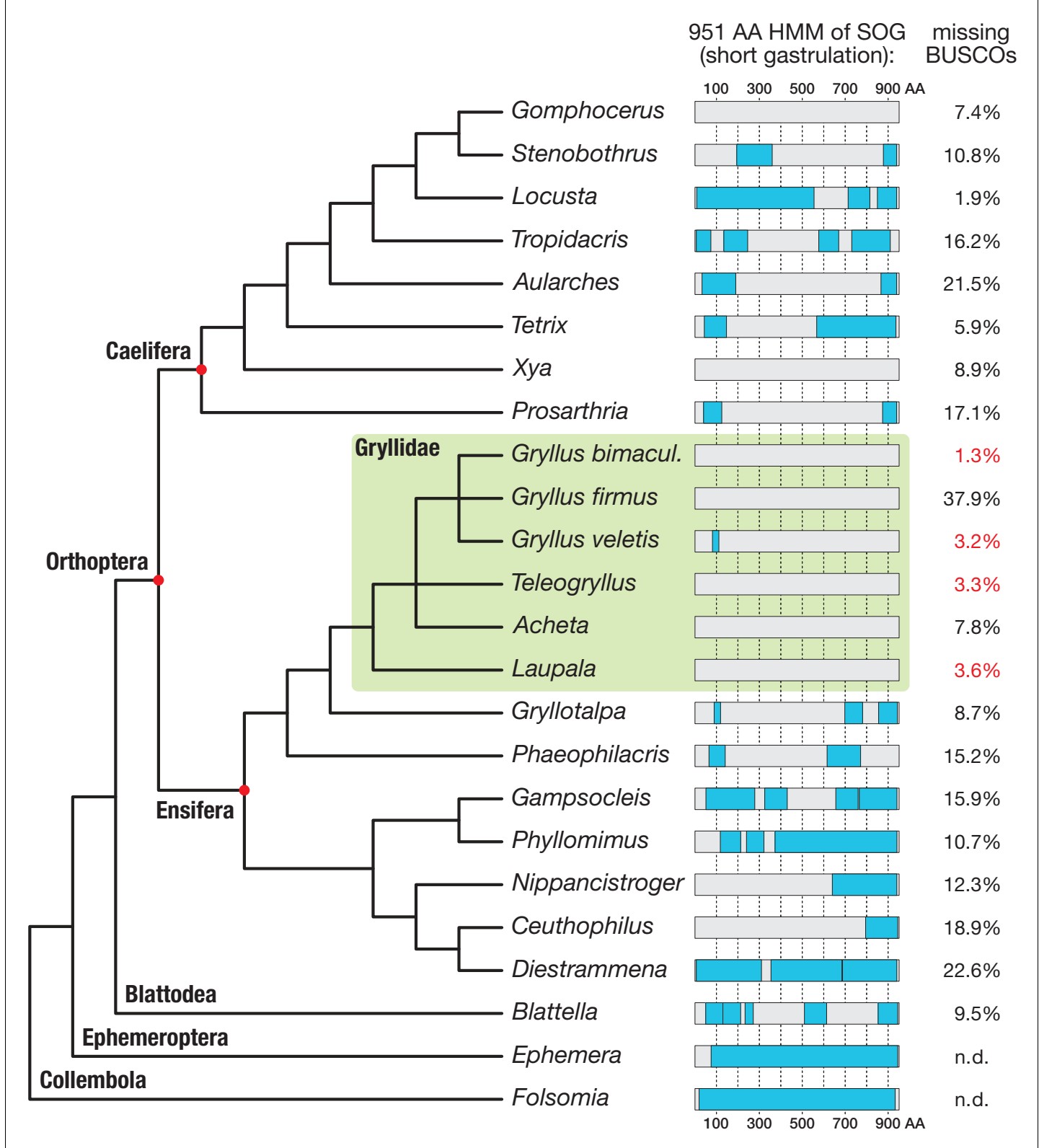

**Figure 2.** Search for short gastrulation (Sog) in Orthoptera. A 951 amino acid long Sog hidden Markov model (HMM, see Materials and methods) was used to identify Sog or Sog fragments in the transcriptomes of 21 orthopteran species and in the outgroup species *Blattella germanica* (*Supplementary file 1* for full species names). Known *Ephemera danica* and *Folsomia candida* Sog sequences were directly scanned with the HMM as control. Sog HMM regions with transcriptomic evidence (i.e. Blast reciprocal best hit-controlled matches to the Sog HMM) are indicated as blue boxes, *Figure 2 continued on next page*

*Figure 2 continued*

mapped onto a schematic phylogenetic tree of the corresponding species (phylogeny after *Leubner, 2017*; *Song et al., 2015*; *Zhang et al., 2013*). The BUSCO (Basic Universal Single Copy Orthologue) values (right column; *Supplementary file 1*) provide a measure for the completeness of the respective transcriptomes. Among the most complete transcriptomes are those of four Gryllidae (green box; missing BUSCOs indicated in red). Sog might have been lost in several orthopteran lineages. In particular, none of the transcriptomes of the Gryllidae contained significant matches to the HMM.

The online version of this article includes the following figure supplement(s) for figure 2:

**Figure supplement 1.** Phylogeny of BMP and TGFβ-like ligand class interrelationships across the Metazoa, as determined by Bayesian (*Huelsenbeck and Ronquist, 2001*) methods.
**Figure supplement 2.** TGFβ and BMP receptor molecule interrelationships across the Metazoa, as determined by Bayesian (*Huelsenbeck and Ronquist, 2001*) methods, and rooted at the midpoint.
**Figure supplement 3.** Smad and Dad interrelationships across the Metazoa, as determined by Bayesian (*Huelsenbeck and Ronquist, 2001*) methods.
**Figure supplement 4.** Brinker, Schnurri and SMURF interrelationships as determined by Bayesian (*Huelsenbeck and Ronquist, 2001*) methods.
**Figure supplement 5.** Noggin, Follistatin and Pentagone interrelationships across the Metazoa, as determined using Bayesian (*Huelsenbeck and Ronquist, 2001*) methods.
**Figure supplement 6.** Crossveinless2/BMPER and KCP (Cv2), Twisted Gastrulation/Crossveinless (Tsg) and Tolloid interrelationships across the Metazoa, as determined by Bayesian (*Huelsenbeck and Ronquist, 2001*) methods.
**Figure supplement 7.** DAN class interrelationships across the Metazoa, as determined by Bayesian (*Huelsenbeck and Ronquist, 2001*) methods.
**Figure supplement 8.** *pipe*, *windbeutel* and *nudel* interrelationships across the Metazoa, as determined by Bayesian (*Huelsenbeck and Ronquist, 2001*) methods.
**Figure supplement 9.** *spaetzle* interrelationships across the Metazoa, as determined by Bayesian (*Huelsenbeck and Ronquist, 2001*) methods.
**Figure supplement 10.** *pelle* and *tube* interrelationships across the Metazoa, as determined by Bayesian (*Huelsenbeck and Ronquist, 2001*) methods.
**Figure supplement 11.** *dorsal*, *relish* and *cactus* interrelationships across the Metazoa, as determined by Bayesian (*Huelsenbeck and Ronquist, 2001*) methods.

homolog, Dorsal has three paralogs in *Gryllus* (*Figure 2—figure supplements 10* and *11*, *Supplementary file 5*).

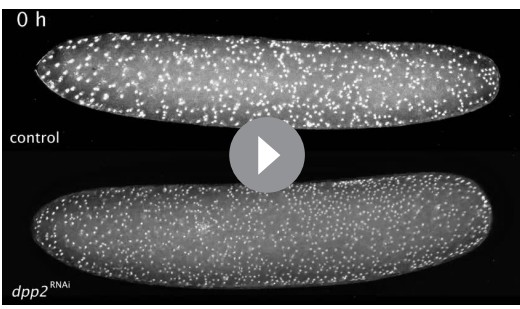

**Video 1.** Early development of an embryo lacking BMP signalling (*Gb-dpp2* knockdown [KD]). Time-lapse imaging using the pXLBGact Histone2B:eGFP transgenic *Gryllus* line (*Nakamura et al., 2010*). The movie shows the development of a control and a *Gb-dpp2* RNAi embryo from uniform blastoderm stage (egg stage 2; embryonic stage 1.5) until embryonic stage 5 (staging according to *Donoughe and Extavour, 2016*). The germ rudiment condenses ventrally in the control embryo (15–30 hr), the serosa closes over the embryo (35–40 hr) and anatrepsis takes place (40 hr onwards). The germ rudiment condenses towards the ventral-posterior in the *Gb-dpp2* KD embryos and ectopic tissue folding is taking place (40 hr onwards). The eggs are oriented with the anterior pointing to the left and ventral down.
https://elifesciences.org/articles/68287#video1

## The pattern of dorsoventral cell types of early *Gryllus* embryos

The orientation of the body axes of *Gryllus* embryos can be readily inferred from the banana shape of the egg: the anterior pole of the embryo is oriented towards the pointed end, the ventral midline towards the convex side of the egg (*Figure 3—figure supplement 1A*; *Sarashina et al., 2005*). Like most insects, *Gryllus* embryos have a syncytial cleavage stage followed by a uniform syncytial and a cellularised blastoderm stage (control embryo in *Video 1*). While most cells of the blastoderm develop into the extraembryonic serosa, cells in a region between 10% and 55% egg length (0% = posterior pole) begin to condense towards the ventral (and mostly posterior) side to form the germ anlage (or germ rudiment comprising the embryo proper and amnion) (*Nakamura et al., 2010*; *Sarashina et al., 2005*). This stage is called differentiated blastoderm. In our analysis, we focus in particular on this stage and the subsequent gastrulation and early germ band elongation stages. During these stages, major subdivisions along the DV axis can be distinguished on the basis of differences in cell density, nuclear size, gene expression and BMP signalling activity.

We monitored differences in cell density and nuclear size either by DNA staining of fixed embryos or by live-imaging using histone2B:eGFP (*Nakamura et al., 2010*). To further distinguish different DV cell types, we performed *in situ* hybridisation (ISH) for *Gb-zerknüllt* (*Gb-zen*), *Gb-SoxN*, *Gb-twist* (*Gb-twi*), *Gb-single minded* (*Gb-sim*) and *Gb-wingless* (*Gb-wg*), as well as antibody staining for phosphorylated Mad (pMad), which provides a read-out of BMP signalling (*Dorfman and Shilo, 2001*). We first describe the wildtype pattern of these markers and subsequently use this information to analyse cell fate shifts observed upon knockdown (KD) of BMP and Toll signalling components.

During early stages of the differentiated blastoderm, the condensing germ anlage consists of two lateral plates of higher cell density separated by more widely spaced ventral cells (*Figure 3A, D*, *Videos 1* and *2*; *Nakamura et al., 2010*; *Sarashina et al., 2005*). The cells of the lateral plates weakly express *Gb-zen* with a relatively sharp expression boundary at the ventral and fading expression towards the dorsal side (*Figure 3A*). In holometabolous insects, *zen* is mostly expressed in extraembryonic tissues (*Buchta et al., 2013*; *Rafiqi et al., 2008*; *van der Zee et al., 2005*). However, in *Oncopeltus*, the only hemimetabolous insect in which *zen* expression has been studied during blastoderm stages, *zen* is also expressed in the condensing germ anlage (*Panfilio et al., 2006*). Similar to *Gb-zen*, *Gb-SoxN* is expressed in the lateral plates during early condensation (*Figure 3D*). This is reminiscent of the expression of *SoxN* in *Drosophila*, which is observed in a broad ventrolateral domain corresponding to the presumptive ventral (neurogenic) ectoderm at the cellular blastoderm stage (*Crémazy et al., 2000*). The ventral cells that are devoid of *Gb-zen* and *Gb-SoxN* expression weakly express *Gb-twi* and *Gb-sim* (*Figure 3—figure supplement 2*). The gene *twist* is expressed in the presumptive mesoderm in all insects studied so far, and functional analyses in *Gryllus* have shown that *Gb-twi* is required for mesoderm specification (*Buchta et al., 2013*; *Donoughe et al., 2014*; *Sachs et al., 2015*). The gene *sim* is required for mesectoderm specification in *Drosophila* (*Thomas et al., 1988*) and is expressed in the mesectoderm of many insects (*Buchta et al., 2013*; *Sachs et al., 2015*; *Stappert et al., 2016*; *Zinzen et al., 2006*). Yet, like in *Gryllus*, early *sim* expression encompasses the entire presumptive mesoderm in *Tribolium*, *Nasonia* and *Oncopeltus*. Together this expression analysis suggests that the lateral plate region with high cell density corresponds to the ectoderm (including in particular the ventral neurogenic ectoderm) and the ventral region with low cell density to the mesoderm of the future embryo.

As the germ anlage undergoes further condensation, the lateral plates move ventrally (*Nakamura et al., 2010*; *Sarashina et al., 2005*). During this stage, *Gb-zen* and *Gb-SoxN* show a complex expression pattern. While *Gb-SoxN* is not expressed in extraembryonic tissues, *Gb-zen* transcription appears in serosa cells in front of the head lobes (white asterisk in *Figure 3B''*). These cells are characterised by widely spaced, large nuclei. Within the germ anlage, *Gb-zen* and *Gb-SoxN* are broadly expressed with higher levels in the future head lobes (*Figure 3B, E*). *Gb-zen* expression evenly extends to the dorsal rim of the head lobes. *Gb-SoxN* expression is more graded with diminishing levels towards the dorsal side. Interestingly, both genes are also upregulated in a stripe of cells straddling the ventral midline. Thus, the expression of *Gb-zen* and *Gb-SoxN* allows for the simultaneous marking of ventral (mesodermal) and lateral (ectodermal) regions of the embryo.

At corresponding stages of germ anlagen condensation, *Gb-twi* is expressed in a sharply demarcated ventral domain that gets wider towards the posterior pole (*Figure 4A*). *Gb-sim* forms two lateral expression stripes abutting the margins of *Gb-twi* expression (*Figure 4D*). These stripes presumably correspond to the future mesectoderm, like in other insects (*Zinzen et al., 2006*). During mesoderm internalisation (*Figure 4—figure supplements 1* and *2*, *Video 2*), the two *Gb-sim* stripe fuse along the ventral midline (*Figure 4E*), and *Gb-twi* expression is turned off except for a small anterior domain (whose anterior margin corresponds to location where the two head lobes meet) and a cup-like region at the posterior pole which is characterised by low cell density (*Figure 4B*). This *Gb-twi* expression profile is reminiscent of *twi* expression in *Tribolium* which decreases in the presumptive mesoderm after internalisation except for expression domains in the head region and at the posterior tip of the embryo (*Handel et al., 2005*).

During early germ band extension, *Gb-zen* is upregulated in most serosa cells and in most ectodermal cells of the embryo except for cells of its dorsal rim (*Figure 3C*). In contrast, *Gb-SoxN* expression becomes restricted to a region in the centre of each head lobe and to a region of the ventral ectoderm, which presumably includes at least the ventral part of the neuroectoderm (*Figure 3F*). *Gb-twi* expression is re-activated in each segment, and its expression at the posterior

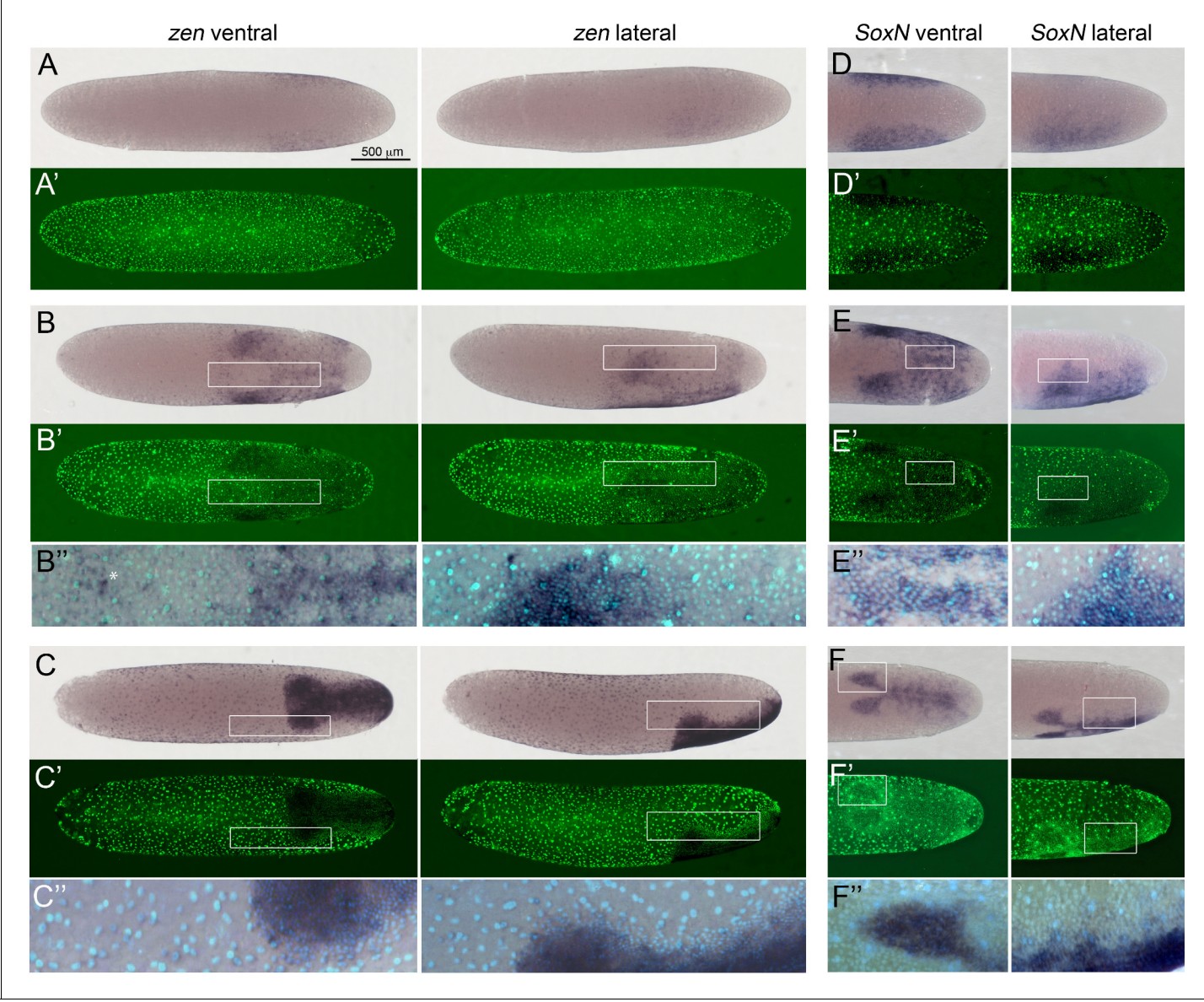

**Figure 3.** Expression of *Gb-zen* and *Gb-SoxN* in early *Gryllus* embryos. (A–C) Whole-mount ISH for *Gb-zen* showing ventral and lateral surface views of embryos (A) at early germ anlage condensation (ES 2.2–2.3), (B) late condensation (ES 2.4–2.5) and (C) early germ band stage (ES 4). (A'–C') DNA staining (Sytox) of respective embryos. (B'', C'') Magnified overlay of ISH and DNA staining of regions boxed in (B, B') and (C, C'). (D, E) Whole-mount ISH for *Gb-SoxN* showing ventral and lateral surface views of the posterior 40% of embryos (D) at early germ anlage condensation (ES 2.2–2.3), (E) late condensation (ES 2.4–2.5) and (F) early band stage (ES 4). (D', E') DNA staining (Sytox) of respective embryos. (E'', F'') Magnified overlay of ISH and DNA staining of regions boxed in (E, E') and (F, F'). All embryos are oriented with the posterior pole pointing to the right. Staging according to *Donoughe and Extavour, 2016*; *Sarashina et al., 2005*. White asterisk in B'' indicates *Gb-zen* expression in serosa cells.

The online version of this article includes the following figure supplement(s) for figure 3:

**Figure supplement 1.** Phenotype classes resulting from reduced Toll and BMP signalling.

**Figure supplement 2.** Expression of (A) *Gb-twi* and (B) *Gb-sim* in early Gryllus embryos.

end of the germ band is maintained (*Figure 4C*; *Kainz, 2009*). *Gb-sim* expression is seen as a single stripe along the ventral midline (*Figure 4F*).

To simultaneously gain information on segmentation and DV patterning, we performed *Gb-wg* ISH (*Miyawaki et al., 2004*). *Gb-wg* expression is initiated in distinct domains of the head lobes and at the posterior margin of the germ anlage during mesoderm internalisation (*Figure 4—figure supplement 2*). As the head lobe expression of *Gb-wg* has well-defined ventral and dorsal boundaries,

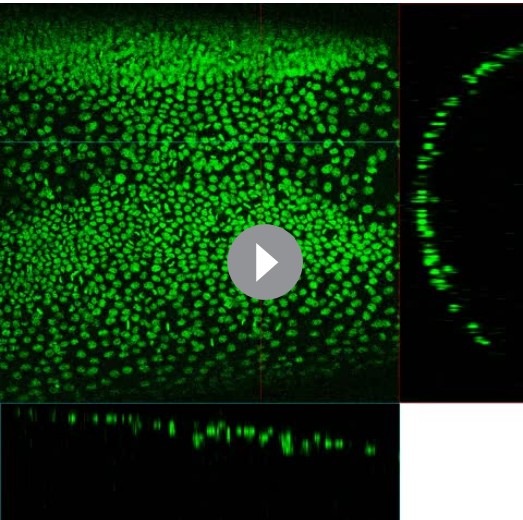

**Video 2.** Mesoderm internalisation. Time-lapse imaging using the pXLBGact Histone2B:eGFP transgenic *Gryllus* line (*Nakamura et al., 2010*). Ventral surface views and z-sections from early to late gem anlage condensation (ES 2.2–ES2.6). Staging according to *Donoughe and Extavour, 2016*; *Sarashina et al., 2005*. The embryo is oriented with the anterior pointing to the left.

https://elifesciences.org/articles/68287#video2

it can be used to detect DV cell fate shifts in addition to the more typical markers described above. By the time mesoderm internalisation is completed in medial positions of the embryo, two anterior *Gb-wg* stripes (antennal and mandibular segments) have formed in addition to the ocular head lobe expression (*Figure 4—figure supplement 2*). During subsequent germ band extension, further *Gb-wg* stripes are added. Early segmental *Gb-wg* expression is restricted to the ventral ectoderm. *Gb-wg* only becomes expressed segmentally in the dorsal ectoderm (*Figure 4—figure supplement 3*) after germ band retraction (stage 11).

The markers presented so far predominantly label ventral cell fates (ventral ectoderm, mesectoderm and mesoderm) in the early embryo, which is the main focus of this study. In order to specifically detect dorsal and dorsolateral cells, we stained blastoderm and early germ band embryos for phosphorylated Mad (pMad), which provides a read-out of BMP signalling (*Dorfman and Shilo, 2001*). High levels of BMP signalling are characteristic for the dorsalmost cell fate, the amnion (or amnioserosa in *Drosophila*) and lower levels for the dorsal (non-neurogenic) ectoderm in all insects studied so far (*Dorfman and Shilo, 2001*; *Özüak et al., 2014b*; *Sachs et al., 2015*; *van der Zee et al., 2006*). The association of BMP activity with (extraembryonic and) non-neurogenic ectoderm holds even outside the insects (*De Robertis, 2008*).

During the early differentiated blastoderm stage (*Figure 5A*), a dorsal to ventral gradient of nuclear localised pMad is present in the region where the condensation of the germ anlage takes place. This indicates that like in other insects a global DV BMP gradient is established in *Gryllus* embryos prior to gastrulation. After the germ anlage condenses towards the ventral side of the egg, the pMad pattern becomes more complex (*Figure 5B*). (1) High levels of pMad are visible in the serosa characterised by widely spaced large nuclei. (2) High levels of pMad are also found in a region with mostly small nuclei surrounding the germ anlage. These cells are closer together than serosa cells, but less densely packed than the remainder of the germ anlage. They are likely to correspond to amnion/dorsal ectoderm. Towards the ventral side of the germ anlage, pMad concentrations sharply decrease and no staining was detected in the densely packed cells that constitute the largest part of the germ anlage (*Figure 5B*). In most insects, low levels of BMP signalling that cannot be detected with pMad antibodies are sufficient to suppress neurogenesis. Thus, the densely packed cells lacking detectable pMad staining are likely to include both non-neurogenic and neurogenic ectoderm.

Using cell density, gene expression and detection of BMP activity, we can distinguish four regions of the differentiated blastoderm in *Gryllus* embryos. (1) Widely spaced cells with large, pMad-positive nuclei that are localised anterior to the condensing germ anlage and express *Gb-zen*. They will give rise to the serosa. (2) pMad-positive cells at the dorsal and anterior rim of the germ anlage that are less densely packed than the remainder of the germ anlage. They are likely to correspond to amnion and (the dorsal rim of) the dorsal ectoderm. (3) pMad-negative, densely packed lateral cells that express *Gb-zen* and *Gb-SoxN* and are ventrally demarcated by a stripe of *Gb-sim* expressing cells. These cells include the neurogenic ectoderm. (4) A ventral domain of cells, which initially are less densely packed than the cells of the lateral plates and express *Gb-twi* and early *Gb-sim*. During condensation, most of these cells lose *Gb-sim* expression, which is maintained only in lateral stripes (presumptive mesectoderm) while the central cells express *Gb-zen* and *Gb-SoxN*. These correspond

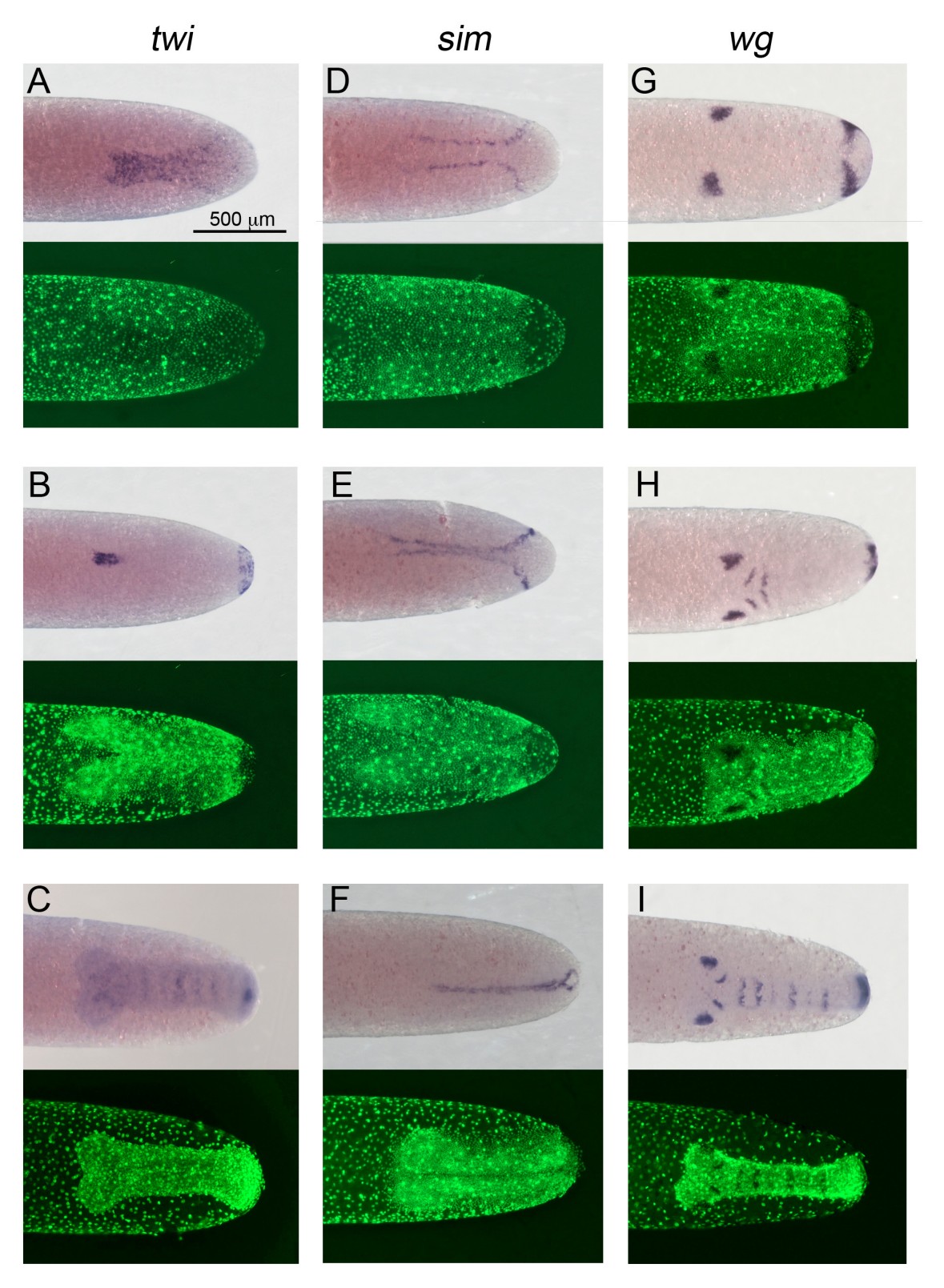

**Figure 4.** Expression of *Gb-twi*, *Gb-sim* and *Gb-wg* in early *Gryllus* embryos. Whole-mount ISH for indicated genes and DNA staining (Sytox) showing ventral surface views of the posterior 40% of embryos at (**A, D**) mid germ anlage condensation (ES 2.4–2.5), (**B, E, G**) late germ anlage condensation (ES 2.6–3.0) and (**C, F, H, I**) during early germ band extension (ES 4.0–4.4). All embryos are oriented with the posterior pole pointing to the right. Staging according to ***Donoughe and Extavour, 2016***; ***Sarashina et al., 2005***.
*Figure 4 continued on next page*

*Figure 4 continued*

The online version of this article includes the following figure supplement(s) for figure 4:

**Figure supplement 1.** Mesoderm internalisation stills from *Video 1*.
**Figure supplement 2.** Mesoderm in early germ band embryos.
**Figure supplement 3.** *wingless* expression in late *Gb-Toll1* knockdown (KD) embryos.

to the presumptive mesoderm, which is internalised when the ectodermal plates close along the ventral midline.

## The requirement for BMP signalling is largely restricted to the dorsal side of *Gryllus* embryos

The early dorsal-to-ventral gradient of pMad (*Figure 5A, B*) suggests that BMP signalling plays an important role in patterning the DV body axis of *Gryllus* embryos. It has been noted before that RNAi with components of the BMP signalling pathway leads to early defects (*Donoughe et al., 2014*). However, the authors of this study focused on weak RNAi phenotypes-producing embryos with a normal DV axis for subsequent analysis of BMP signalling during primordial germ cell development.

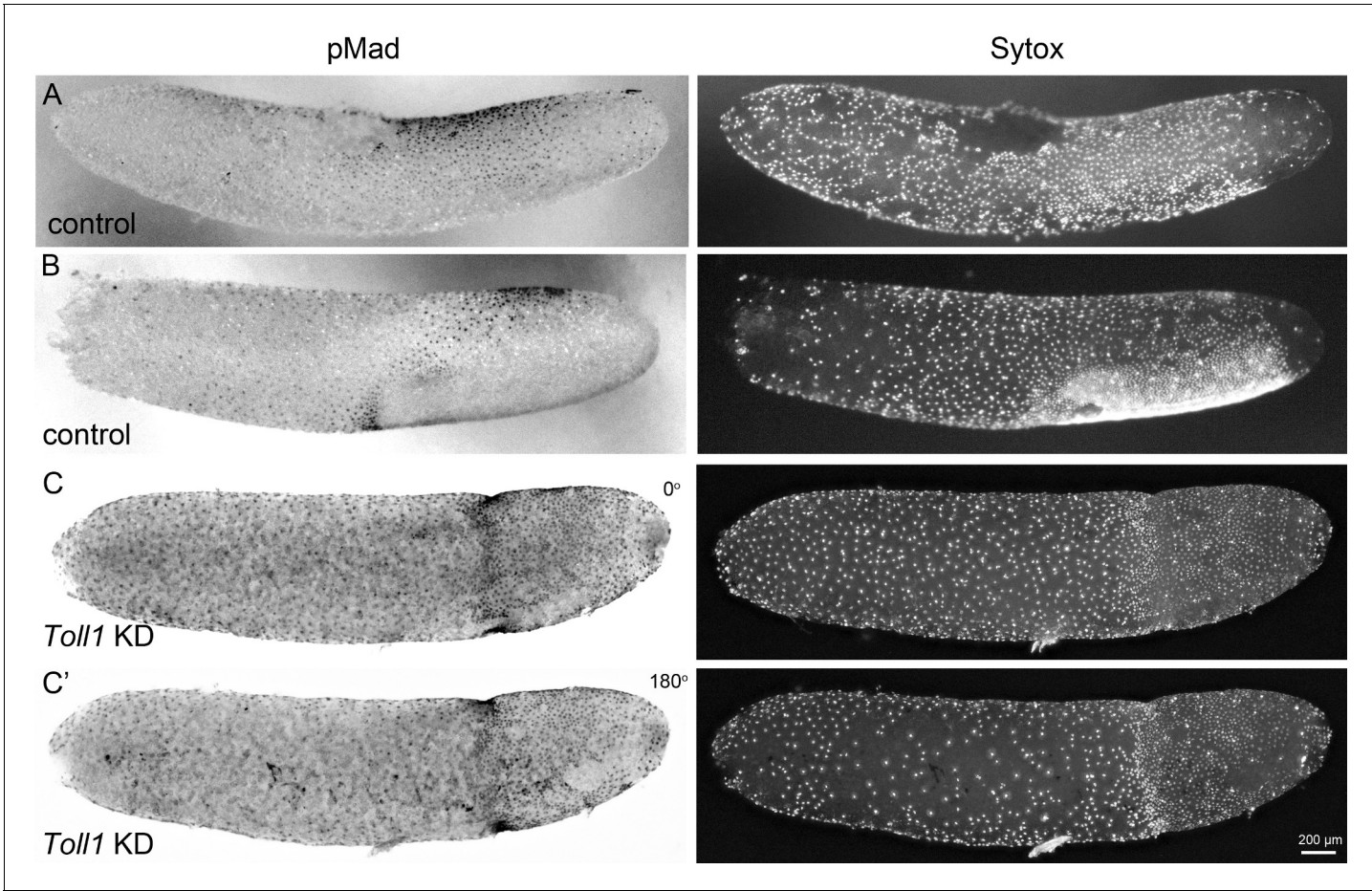

**Figure 5.** BMP pathway activity in early control and *Gb-Toll1* pRNAi embryos. Lateral surface views of embryos stained for phosphorylated Mad (pMad) and DNA (Sytox). The degree of nuclear accumulation of pMad indicates the strength of BMP signalling. (**A**) Control embryo at the beginning of germ anlage condensation (ES 2.0). Nuclear pMad forms a dorsal-to-ventral gradient between 20% and 60% egg length (0% is the posterior pole). (**B**) Control embryo at late germ anlage condensation (ES 2.6–3.0). Cells with high nuclear pMad levels surround the condensing germ anlage. (**C**) *Gb-Toll1* knockdown embryo showing a radially symmetric distribution of nuclear pMad. The embryo is in a similar stage than the control embryo shown in (**B**). (**C'**) The embryo shown in (**C**) turned around by 180°. Staging according to *Donoughe and Extavour, 2016*; *Sarashina et al., 2005*.

To study the role of BMP in axis formation, we knocked down the three BMP ligands, *Gb-dpp1*, *Gb-dpp2* and *Gb-gbb*, and the BMP receptor *Gb-tkv* (*Figure 3—figure supplement 1C, D*). Parental RNAi with *Gb-dpp1* led to sterility of the injected females (n < 10). For this reason, we failed to analyse the effect of *Gb-dpp1* on early *Gryllus* embryogenesis. KD of *Gb-gbb* and *Gb-tkv* led to severe, but incomplete, sterility, allowing the collection of a small number of eggs for phenotypic analysis. No sterility was observed for parental RNAi with *Gb-dpp2*. To quantify the strength of the KDs, we analysed the frequency of late (stage 11) phenotypes as they could be easily classified on the basis of simple morphological criteria (*Figure 3—figure supplement 1C, D*). This analysis indicates that strong phenotypes were obtained for KDs of all three genes (*Gb-tkv*, *Gb-dpp2* and *Gb-gbb*), albeit with the highest frequency for KD of *Gb-tkv*.

To get a first impression of the development of early embryos with compromised BMP signalling, we performed time-lapse imaging. Embryos collected from pXLBGact Histone2B:eGFP females injected with *Gb-dpp2* or *Gb-tkv* dsRNA showed that germ anlage condensation did not result in two distinct lateral plates with high cell density (*Figure 6*, *Videos 1*, *3* and *4*). Rather, condensation was mainly directed towards the posterior pole and led to the formation of densely packed cells also at the dorsal side of the condensation zone. However, at the ventral side we observed a region of less densely packed cells corresponding to the mesodermal domain in control embryos. During further condensation, this region disappeared, presumably through internalisation of the mesoderm. These embryos were composed of only densely packed cells and in extreme cases were almost rotationally symmetric (*Figure 6—figure supplement 1*). Together, these observations indicate that interfering with BMP signalling leads to a loss of dorsal parts of germ anlage (presumably presumptive amnion and dorsal ectoderm) and a compensatory expansion of the ventral ectodermal regions. However, loss of BMP does not prevent the formation of a distinct mesodermal domain.

The phenotypic analysis based on live imaging is supported by the changes of marker gene expression upon KD for BMP signalling components. As the strongest phenotypes were observed after *Gb-tkv* KD, we have collected the most complete dataset for *Gb-tkv* RNAi. However, similar results were obtained for KD of *Gb-dpp2* and *Gb-gbb* (*Figure 7—figure supplement 1*). Shortly before gastrulation, *Gb-zen* is expressed in the head lobes and a ventral stripe straddling the ventral midline in control embryos (*Figure 3B* and *Figure 7A1*). After KD of *Gb-tkv*, the ventral gap between the two head lobe domains increased and their dorsal margins were shifted to the dorsal side so that a solid expression band forms stretching over the dorsal midline (*Figure 7—figure supplement 2*). Interestingly, the ventral midline expression is maintained and was only slightly wider than in controls (*Figure 7—figure supplement 2*). Similar expression changes are observed for *Gb-SoxN* in embryos of corresponding stages: head lobe expression expands to the dorsal side, while mesodermal expression remains restricted to the ventral side (*Figure 7B2*). After gastrulation, *Gb-SoxN* is upregulated in the centre of the head lobes and in ventral parts of the ectoderm in control embryos. Upon *Gb-tkv* KD, the head lobe domain expands to the dorsal midline, while the ventral domain appears only slightly broader than in control embryos (*Figure 7C, C'*). However, since KD embryos show reduced degrees of germ band extension, which in control embryos leads to an elongation of the AP and a narrowing of the DV axis, it is not clear whether the broadening of the ventral *Gb-SoxN* domain is due to cell fate changes or altered morphogenesis. The restricted ventral *Gb-SoxN* expression nevertheless indicates that the specification of ventral cell fates, including mesoderm and the ventralmost parts of the ectoderm, is not drastically altered upon loss of BMP signalling in *Gryllus*. This conclusion is further supported by the analysis of *Gb-twi*, *Gb-sim* and *Gb-wg* expression. While it was very difficult to obtain reliable data of KD embryos for the earliest stages of *Gb-twi* expression, gastrulation-stage embryos were recovered which harbour an anterior *Gb-twi* domain. Compared to controls, this domain is shorter and laterally expanded, but still remains clearly confined to the ventral side (*Figure 7D*, black arrows). Like for *Gb-SoxN*, simultaneous AP shortening and DV expansion of the anterior *Gb-twi* might be due to a lack of convergent extension of the KD embryos rather than to DV fate shifts. A pattern similar to controls was also observed for *Gb-sim* in KD embryos: *Gb-sim* refined into two stripes which fused, albeit irregularly, along the ventral midline (*Figure 7E*).

Finally, although in KD embryos the ventral distance between the ocular domains of *Gb-wg* expression increased and expanded to the dorsal midline (*Figure 7F*), the subsequent formation of more posterior *Gb-wg* stripes revealed a clearly defined DV polarity within the ectoderm with *Gb-wg* stripe formation being restricted to a narrow ventral domain (*Figure 7G*). Together these

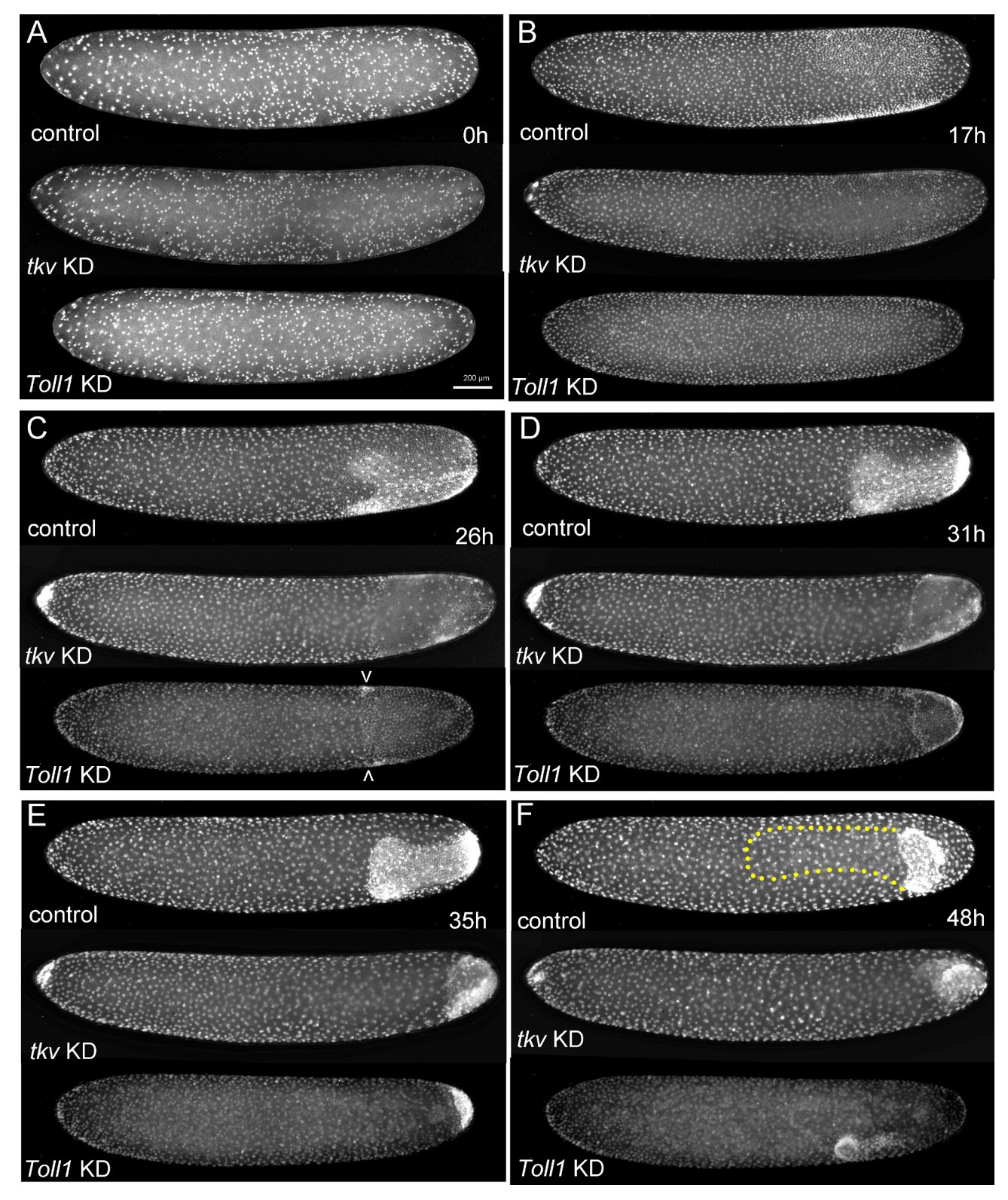

**Figure 6.** The development of *Gryllus* embryos lacking Toll or BMP signalling. Stills from videos of embryos which express Histone2B:eGFP under an ubiquitously active promoter (pXLBGact Histone2B:eGFP) (*Nakamura et al., 2010*). Control embryos are compared to embryos of corresponding developmental stages resulting from pRNAi knockdown (KD) *Gb-tkv* or *Gb-Toll1*. (**A**) Uniform blastoderm (ES 1.6). (**B**) Early germ anlagen condensation (ES 2.2–2.3). (**C**) Late germ anlage condensation (ES 2.6). (**D**) Early germ band stage (ES 3.0). (**E**) Early germ band elongation (ES 4.0–4.3). (**F**) Early

*Figure 6 continued on next page*

Figure 6 continued

anatrepsis (ES 5.0). The position of the germ band of the control embryo is marked by a yellow dotted line. Staging according to *Donoughe and Extavour, 2016*; *Sarashina et al., 2005*.

The online version of this article includes the following figure supplement(s) for figure 6:

**Figure supplement 1.** Cell densities in embryos lacking Toll or BMP signalling.

observations demonstrate that BMP signalling is required for patterning of the dorsal ectoderm/amnion in *Gryllus,* but has only a weak, potentially secondary, influence on ventral ectoderm and mesoderm (for later development, see *Figure 7—figure supplements 3* and *4*).

## Toll signalling is required along the entire DV axis of *Gryllus* embryos

Our functional analysis of BMP signalling in *Gryllus* indicates that the specification and patterning of ventral cell types is controlled by BMP-independent signalling mechanisms. Since Toll signalling induces ventral cell types in all other insects, we analysed the function of this pathway by downregulating candidates for the receptor Toll and the downstream transcription factor Dorsal.

From a previous study, we knew that there is a single *Gb-Toll1* ortholog present in *G. bimaculatus* (*Benton et al., 2016*). We cloned three non-overlapping fragments of the gene (5′, mid and 3′) and analysed the function of *Gb-Toll1* via parental RNAi. We observed a very strong phenotype for all three fragments (*Figure 3—figure supplement 1A, B*).

Time-lapse imaging of embryos collected from pXLBGact Histone2B:eGFP females injected with *Gb-Toll1* dsRNA did not reveal developmental abnormalities before germ anlagen condensation (*Figure 6*, *Video 5*). However, during condensation, no modulation of cell densities along the DV axis was observed. In particular, the formation of distinct lateral plates was absent. Instead, the posterior third of cells condensed towards the posterior pole in a radially symmetric way. This process resulted in a small posterior cap of cells sharply demarcated from the serosa cells (*Figure 6E*, *Video 5*). However, unlike loss of BMP signalling, the loss of Toll leads to embryos with lower cell densities, presumably corresponding to dorsal regions of control embryos (*Figure 6—figure supplement 1*). This suggests that Toll KD embryos consist only of amnion/dorsal ectoderm.

Marker gene expression in KD embryos supports this view. Thus, the almost radially symmetric germ anlagen of KD embryos uniformly expressed low levels of *Gb-zen* at differentiated blastoderm stages (*Figure 8A*) and high levels at stages corresponding to germ band extension when the embryo has condensed to form a small, circular disc at the posterior pole (*Figure 9A*). KD embryos lacked *Gb-SoxN* expression (*Figure 8B*). We assume that the difference to *Gb-zen* is due to the fact that *Gb-SoxN* expression does not fully extend to the dorsal margin of the head lobes in control embryos (*Figure 3B, E*).

*Gb-twi* was only expressed in a posterior symmetric cap (*Figure 8C*, *Figure 9C*) after Toll KD. A similar, apparently Toll-independent, *twi* expression domain at the posterior tip of the embryo is also found in *Tribolium* and *Oncopeltus* (*Nunes da Fonseca et al., 2008*; *Sachs et al., 2015*). *Gb-sim* expression was also absent in KD embryos, except for a small symmetric ring at the posterior pole (*Figure 9B*). The same holds for *Gb-wg,* which was only expressed in a posterior symmetric ring (*Figure 9D*). During stages corresponding to germ band extension in control embryos, no *Gb-wg* stripes were observed. This does not exclude that segmentation takes place in *Gb-Toll1* KD embryos as the *Gb-wg* stripes form only within the ventral ectoderm. Thus, loss of *Gb-wg* stripes only reflects the loss of ventral cell fates. This idea is confirmed by analysing late developmental stages (stage 11) when *Gb-wg* is expressed in dorsal parts of each

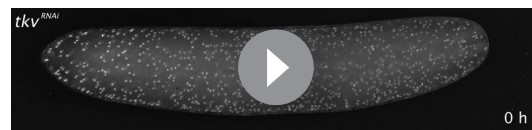

**Video 3.** Complete development of embryo lacking BMP signalling (*Gb-tkv* knockdown [KD]). Six days of constant live imaging using the pXLBGact Histone2B:eGFP transgenic *Gryllus* line (*Nakamura et al., 2010*). The movie shows the development of a *Gb-tkv* RNAi embryo from uniform blastoderm stage (egg stage 2, staging according to *Donoughe and Extavour, 2016*) until embryonic stage 11. The malformed *Gb-tkv* RNAi embryo is released from the yolk and the serosa at the end of the movie (132 hr onwards). The egg is oriented with the anterior pointing to the left and ventral down.
https://elifesciences.org/articles/68287#video3

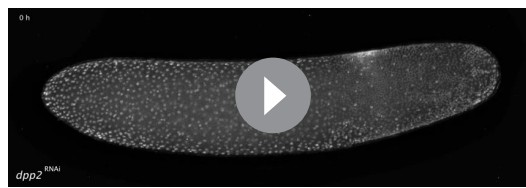

**Video 4.** Early development of embryo lacking BMP signalling (*Gb-dpp2* knockdown [KD]). Time-lapse imaging using the pXLBGact Histone2B:eGFP transgenic *Gryllus* line (*Nakamura et al., 2010*). The movie shows the development of a *Gb-dpp2* RNAi embryo from early germ anlage condensation (ES 2.2) until ES 5. The border between serosa and germ anlage lacks dorsoventral asymmetry. The germ anlage condenses symmetrically towards the posterior pole. Staging according to *Donoughe and Extavour, 2016*; *Sarashina et al., 2005*.

https://elifesciences.org/articles/68287#video4

segment. At this stage, *Gb-Toll1* KD embryos consist of an irregular tissue cluster internalised into the yolk (*Video 6*, *Figure 4—figure supplement 3*). This cluster contains elongated embryonic fragments with *Gb-wg* stripes, indicating that they have dorsal ectodermal identity. These results also show that the segmentation process does not depend on establishing DV polarity, like in other insects. Taken together, the expression of multiple marker genes demonstrates that, in the absence of Toll signalling, *Gryllus* embryos are not able to establish embryonic DV polarity and that they lack all ventral and lateral cell fates being uniformly dorsalised.

We wondered how BMP activity was distributed in these embryos. In contrast to control embryos, pMad staining lacked DV asymmetry in Toll KD embryos and nuclei with the same pMad level were found along the entire DV axis (*Figure 5*). The highest pMad levels were found in a radially symmetric ring of anterior germ anlagen cells abutting the serosa. The serosa cells anterior to the ring showed high levels of pMad, and the germ anlage cells posterior to the ring medium levels. These latter levels correspond to the dorsolateral regions of control embryos. This suggests that upon loss of Toll the majority of embryonic cells do not acquire dorsalmost cell fates (amnion), but are rather dorsalised at the level of the dorsal ectoderm. This is in line with the observation that late *Gb-Toll1 KD* embryos form segments, which express *Gb-wg* and are thus likely composed of dorsal ectoderm. Likewise, *Toll* mutant *Drosophila* embryos do not consist of amnioserosa (the dorsalmost cell fates), but rather of dorsal ectoderm. Together our data demonstrate that *Gryllus* embryos require the Toll pathway for all aspects of DV polarity including the polarisation of BMP signalling.

Toll signalling acts via the NF-κB transcription factor Dorsal (*Stein and Stevens, 2014*). From our transcriptome, we identified three paralogous *dorsal* genes that group with *dorsal* orthologs of many insect species (*Figure 2—figure supplement 11*). However, only *Gb-dl1* resulted in a similar phenotype to that of *Gb-Toll1* KD (*Figure 3—figure supplement 1*). Time-lapse imaging of embryos collected from pXLBGact Histone2B:eGFP females injected with *Gb-dl1* dsRNA reveals that also the KD of *Gb-dl1* is leading to a phenotype in which posterior located cells condense towards the posterior of the egg in a radially symmetric fashion (*Figure 9—figure supplement 1*). In contrast to *Gb-Toll1* RNAi, we find some mildly affected *Gb-dl1* RNAi embryos that emerged from the yolk during katatrepsis and developed into larvae that were close to hatching (*Figure 3—figure supplement 1*). These larvae showed constrictions of dorsalised cuticle and ventral fusions of appendages and eyes (*Figure 7—figure supplement 3A–D*). The ventral fusion of eyes and appendages indicates that reduced *Gb-dl1* amounts lead to a loss of ventral and a compensating expansion of dorsal ectodermal regions. Upon KD of Dorsal or other Toll signalling components in other insects except for *Drosophila*, we have not observed weak phenotypes, which would result in defined DV shifts of adult structures. In *Drosophila*, the observation of different degrees of dorsalisation, which could be arranged in a phenotypic series, provided the first evidence for a morphogen gradient mechanism (*Anderson et al., 1985b*; *Nüsslein-Volhard et al., 1980*; *Roth et al., 1991*).

## Ovarian polarisation of Toll signalling

In *Drosophila*, Toll signalling is initiated by binding of activated Spätzle protein to the Toll receptor (*Stein and Stevens, 2014*). Spätzle protein is produced by the embryo and secreted into the perivitelline space as inactive pro-form. The cleavage of pro-Spätzle depends on a protease cascade that is active in the perivitelline space at the ventral side of the egg. To investigate whether the activation of Toll in *Gryllus* occurs via a similar mechanism, we performed RNAi for potential *Gryllus spz* homologs (*Figure 2—figure supplement 9*). The KD of *Gb-spz2* resulted in the typical late phenotype, which was also observed for KD of *Gb-Toll1* and *Gb-dl1* (*Figure 3—figure supplement 1*). This

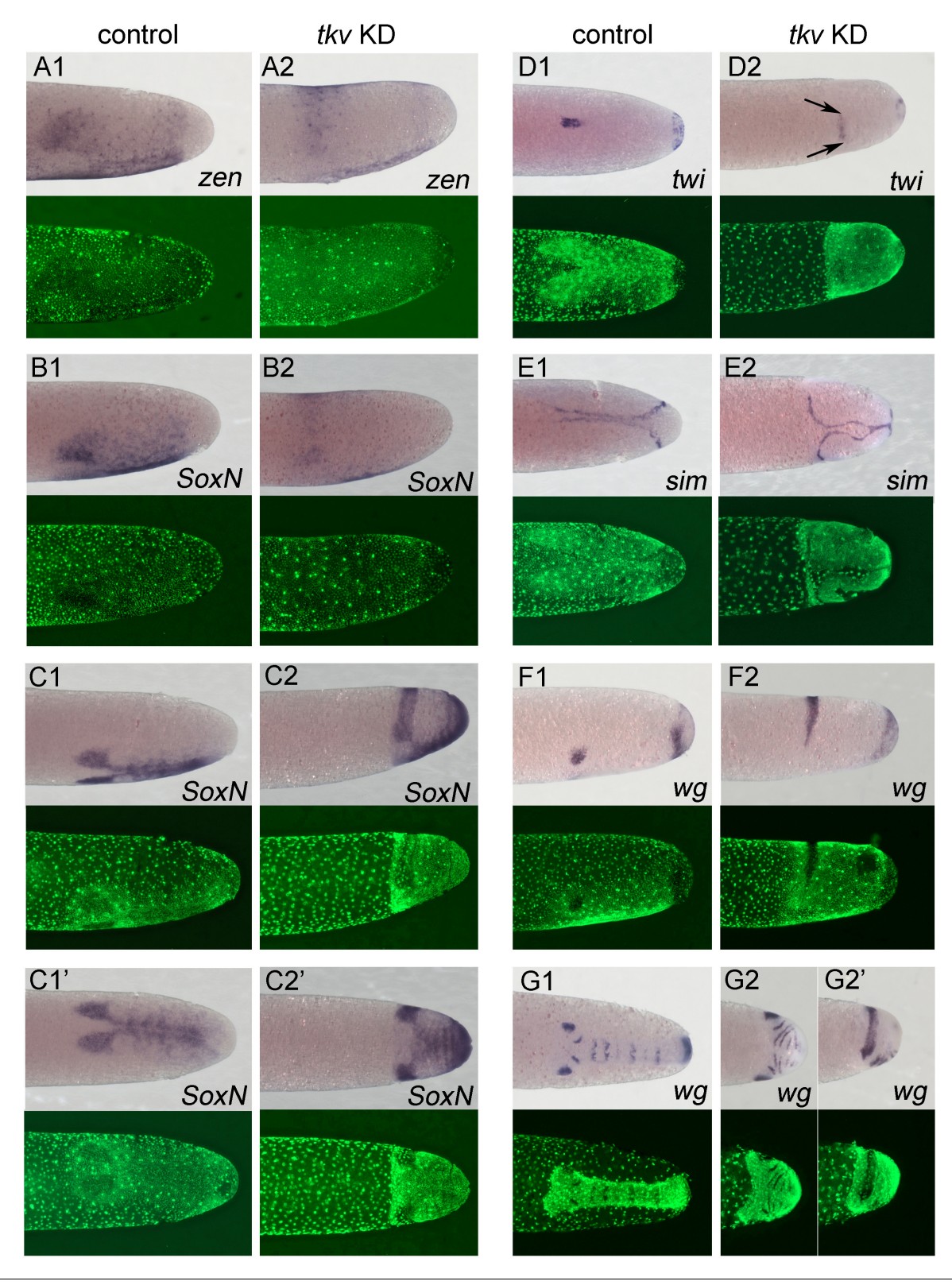

**Figure 7.** Altered dorsoventral (DV) fate map upon loss of BMP signalling. Whole-mount ISH (purple) and DNA staining (green). Only the posterior part of each embryo harbouring the germ anlage is shown. (A1–G1) Control embryos. (A2–G2) *Gb-tkv* knockdown (KD) (pRNAi) embryos. (A1, A2) Lateral surface views of embryos at late germ anlage condensation showing *Gb-zen* expression (ES 2.4–2.5). (B1, B2) Lateral surface views of embryos at late germ anlage condensation showing *Gb-SoxN* expression (ES 2.4–2.5). (C1, C2) Lateral and (C1', C2') ventral surface views of embryos at early

*Figure 7 continued on next page*

*Figure 7 continued*

elongating germ band stages (ES 4) showing *Gb-SoxN* expression. (D1, D2) Ventral surface views of embryos at early germ band stages (ES 3.0) showing *Gb-twi* expression. Two black arrows mark the extent of *Gb-twi* expression upon *Gb-tkv* KD. (E1, E2) Ventral surface views of embryos at early germ band stages (ES 3.0) showing *Gb-sim* expression. (F1, F2) Lateral surface views of embryos at late germ anlage condensation showing *Gb-wg* expression (ES 2.4–2.5). (G1, G2) Ventral surface views of embryos after formation of segment T3 (ES 4.3) showing *Gb-wg* expression. (G2') Lateral surface view of the *Gb-tkv* KD embryo shown in (G2). Staging according to *Donoughe and Extavour, 2016*; *Sarashina et al., 2005*.

The online version of this article includes the following figure supplement(s) for figure 7:

**Figure supplement 1.** Comparison of dorsoventral fate map shifts after *Gb-tkv*, *Gb-dpp2* and *Gb-gbb* knockdown.
**Figure supplement 2.** *Gb-zen* expression upon *Gb-tkv* knockdown (KD).
**Figure supplement 3.** Late phenotypes produced by interfering with Toll and BMP signalling.
**Figure supplement 4.** The development of *Gb-dpp2* knockdown (KD) embryos.

indicates that also in *Gryllus* Toll is activated by a secreted extraembryonic signal which is restricted to the ventral portion of the egg.

In *Drosophila*, ventral regions of the inner eggshell (vitelline membrane) contain sulfated vitelline membrane proteins, which are required together with the uniformly expressed vitelline membrane protease Nudel (Ndl) to initiate the protease cascade leading to the cleavage and activation of Spz (*Hong and Hashimoto, 1995*; *Sen et al., 1998*; *Stein and Stevens, 2014*). The region-specific sulfatation of these vitelline proteins depends on the sulfotransferase Pipe expressed in the ventral follicular epithelium, the cell layer producing the eggshell (*Sen et al., 1998*). A direct involvement of *pipe* in DV patterning has so far only been observed in *Drosophila* (*Sen et al., 1998*) and *Oncopeltus* (*Chen, 2015*). In *Drosophila*, *pipe* shows a strongly asymmetric expression along the DV axis of the egg chamber and thereby prefigures the region of the egg where Toll signalling will be activated upon egg deposition (*Figure 10—figure supplement 1B*). Neither in *Tribolium* nor in *Nasonia pipe* has been found to be expressed in the follicular epithelium, indicating that major groups of the holometabolous insects (Coleoptera and Hymenoptera) use a mechanism different from *Drosophila* for transmitting DV polarity from the eggshell to the embryo (*Figure 10—figure supplement 1B*). In *Oncopeltus*, *pipe* is expressed evenly along the DV axis of the follicular epithelium or it has only a weak asymmetry, which we could not reliably detect (*Figure 10—figure supplement 1B*). *Oncopeltus* DV patterning is largely a self-organising process in which Toll provides only a weak polarity cue (*Sachs et al., 2015*). Thus, it does not require a high degree of spatial information from the eggshell.

The *Gryllus* genome contains *ndl* and *pipe* homologs (*Figure 2—figure supplement 8*). We wondered whether *Gb-ndl* and *Gb-pipe* were expressed in *Gryllus* ovaries. The *Gryllus* ovary represents the ancestral panoistic type of oogenesis characterised by a lack of nurse cells (*Lynch et al., 2010*). Compared to *Gryllus*, all the above-mentioned insects have advanced

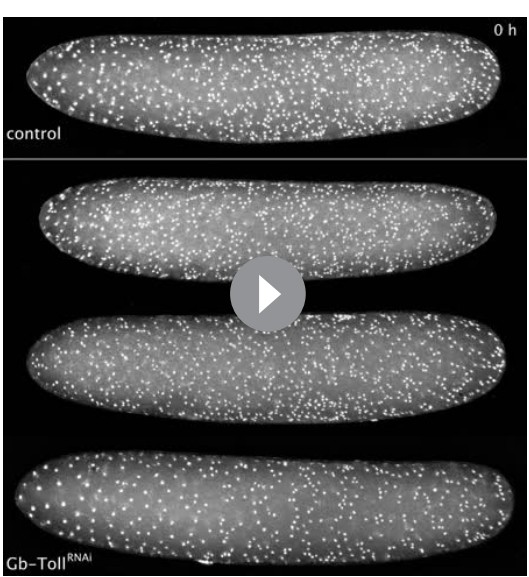

**Video 5.** Development of embryos lacking Toll signalling (*Gb-Toll1* knockdown [KD]). Time-lapse imaging using the pXLBGact Histone2B:eGFP transgenic *Gryllus* line (*Nakamura et al., 2010*). The movie shows the development of a control and three *Gb-Toll1* RNAi embryos from uniform blastoderm stage (egg stage 2; embryonic stage 1.5) until embryonic stage 6–7 (staging according to *Donoughe and Extavour, 2016*). The germ rudiment condenses ventrally in the control embryo (10–30 hr), the serosa closes over the embryo (30–40 hr) and anatrepsis takes place (35 hr onwards). The germ rudiment condenses towards the posterior in the *Gb-Toll1* KD embryos (20–35 hr). However, the posterior cap of cells sinks into the yolk during anatrepsis (35 hr onwards). The eggs are oriented with the anterior pointing to the left and ventral down.
https://elifesciences.org/articles/68287#video5

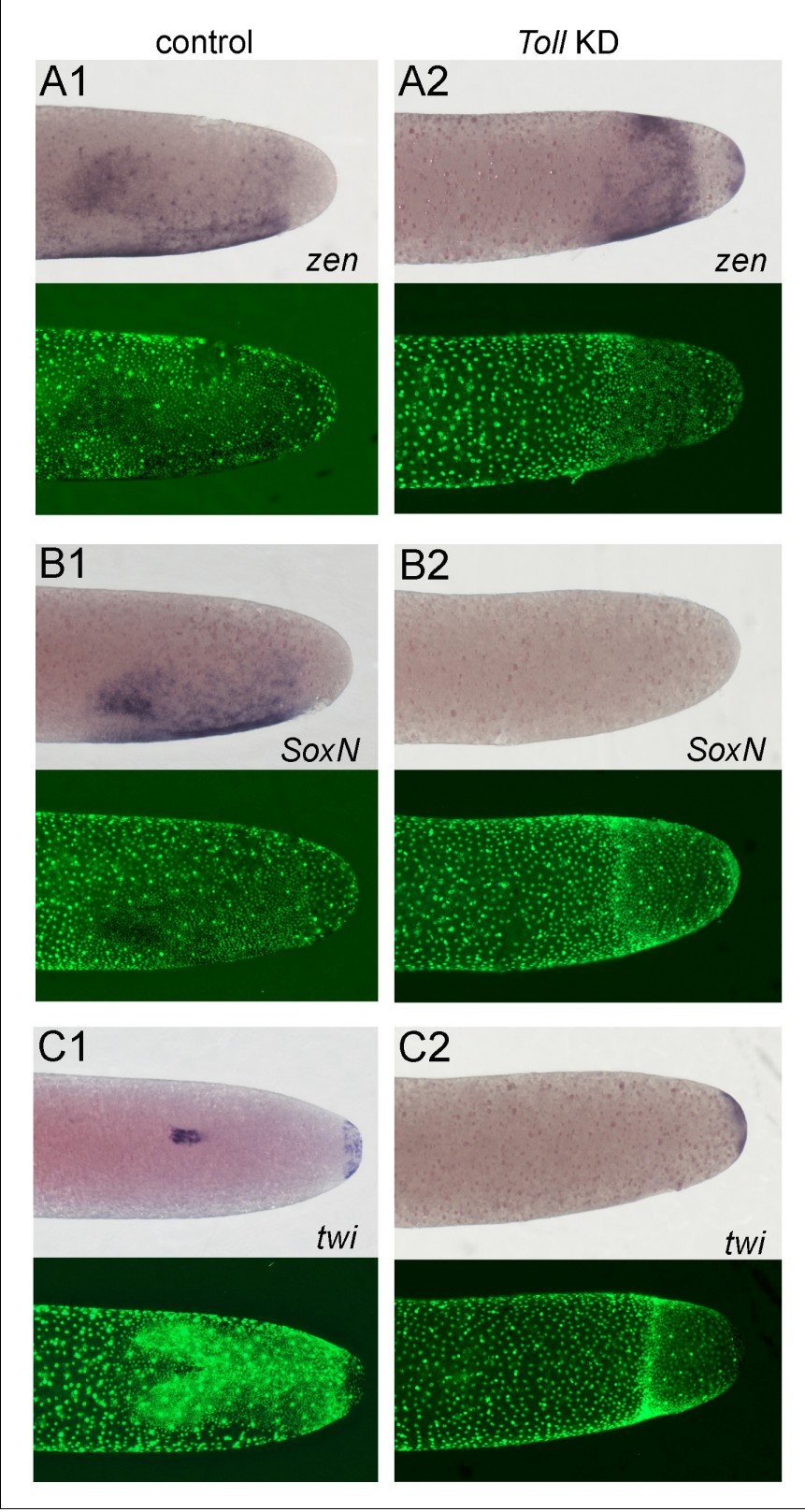

**Figure 8.** Altered dorsoventral (DV) fate map of early embryos lacking Toll signalling. Whole-mount ISH (purple) and DNA staining (green). Only the posterior part of each embryo harbouring the germ anlage is shown. (**A1–C1**) Control embryos. (**A2–C2**) *Gb-Toll1* knockdown (KD) (pRNAi) embryos. (**A1, A2**) Lateral surface views of embryos at late germ anlage condensation showing *Gb-zen* expression (ES 2.4–2.5). (**B1, B2**) Lateral surface views of embryos

*Figure 8 continued on next page*

*Figure 8 continued*

at late germ anlage condensation showing *Gb-SoxN* expression (ES 2.4–2.5). (**C1, C2**) Ventral surface views of embryos at early germ band stages (ES 3.0) showing *Gb-twi* expression. Staging according to *Donoughe and Extavour, 2016*; *Sarashina et al., 2005*.

ovaries with nurse cells (meroistic), either of the telotrophic (*Tribolium*, *Oncopeltus*) or polytrophic subtype (*Nasonia*, *Drosophila*) (*Lynch et al., 2010*). Due to its panoistic type, the ovary of *Gryllus* has a simple architecture (*Figure 10—figure supplement 1A*). The egg chambers are only composed of an oocyte surrounded by a simple monolayer of one type of follicle cells. The growing egg chambers assume the banana shape of the later egg when the oocyte nucleus moves to one side of the oocyte, the future dorsal side of the egg and embryo (*Figure 10—figure supplement 1A*). At this stage, *Gb-ndl* begins to be expressed in the follicular epithelium and assumes high levels of expression in all follicle cells of maturing egg chambers (*Figure 10—figure supplement 1A, B*).

Gb-pipe expression starts evenly in the follicular epithelium prior to the asymmetric movement of the oocyte nucleus. However, after oocyte nucleus movement *Gb-pipe* expression becomes highly asymmetric in the follicular epithelium (*Figure 10*, *Figure 10—figure supplement 1A, B*). Strong expression is seen in anterior, posterior and ventral follicle cells, while *Gb-pipe* is repressed in a broad dorsal-to-lateral domain of the follicle. The oocyte nucleus is localised at the dorsal centre of this repression domain (arrow in *Figure 10A* and section shown in *Figure 10A\*–A\*\*\**). Like in other insects, EGFR signalling is activated in the follicle cells close to the oocyte nucleus of *Gryllus* ovaries and it is required to induce DV polarity in the egg chamber (*Lynch et al., 2010*).

We wondered whether EGF signalling in *Gryllus* is, like in *Drosophila*, a negative regulator of *pipe* expression. We therefore analysed *Gb-pipe* expression upon KD of the EGFR ligand *Gb-Tgfα*. Reduction of EFG signalling leads to a de-repression of *Gb-pipe*, which becomes activated almost evenly in all follicle cells (*Figure 10B, C*). These observations indicate that *Gryllus* uses EGF signalling like *Drosophila* to establish a pronounced DV asymmetry of *pipe* expression in the follicular epithelium. Due to sterility after *Gb-pipe* KD, we were not able to functionally demonstrate that *Gb-pipe* is required for embryonic DV patterning. However, given its expression pattern, its regulation by EGF and the observed dependence of Toll activation on *Gb-spz*, it is very suggestive that in *Gryllus* like in *Drosophila* *pipe* expression prefigures the region of Toll activation and consequently the embryonic region of mesoderm and neuroectoderm formation. Together our results provide evidence for an unexpected similarity between DV patterning of insects (*Drosophila* and *Gryllus*) belonging to distant branches of the phylogenetic tree and representing distinct modes of embryonic and ovarian development.

## Discussion

To understand how the Toll and BMP pathways have been variously deployed across the insects, here we investigated the regulation of DV patterning in *Gryllus*. We find strong similarities between *Gryllus* and *Drosophila*, which is surprising given their evolutionary distance and the alternative strategies employed in other species (*Figures 1* and *11*). A number of molecular and phenotypic observations in the current work support an evolutionary scenario of convergent evolution between crickets and flies, which we present here and discuss in terms of implications for gene regulatory network (GRN) logic and species-specific developmental constraints.

### Similarities between *Gryllus* and *Drosophila*

The observed parallels between *Gryllus* and *Drosophila* concern both embryonic and ovarian aspects of DV axis formation. While the embryonic features are also shared with *Tribolium*, the ovarian ones are in common with *Drosophila* only.

First, the requirement of BMP signalling in *Gryllus* appears to be restricted to the dorsal side of the embryo like in *Drosophila* and *Tribolium* (*Arora and Nusslein-Volhrad, 1992*; *Nunes da Fonseca et al., 2010*; *Ray et al., 1991*; *van der Zee et al., 2006*). The loss of BMP signalling in *Gryllus* has dramatic consequences for embryonic development as the embryo does not condense to the ventral side and axial elongation by convergent extension is strongly compromised (*Figure 6D*). The dorsal half of the embryo lacks DV polarity, as exemplified by the even expansion of the *Gb-SoxN*

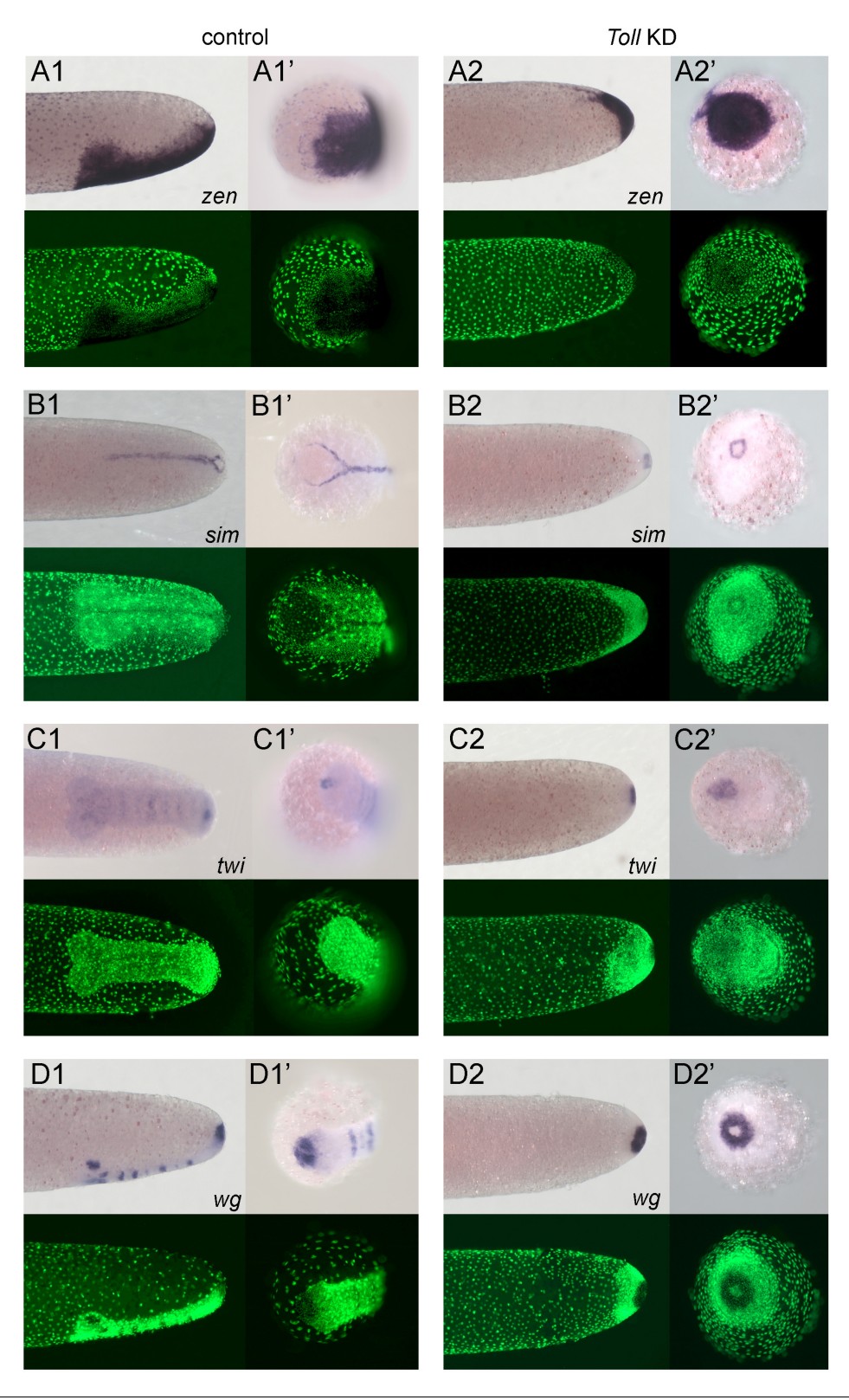

**Figure 9.** Embryos lacking Toll signalling are rotationally symmetric. Whole-mount ISH (purple) for indicated genes and DNA staining (green). (A1–D1, A2–D2) Posterior 40% of embryos at early germ band stages (ES 4.0–4.3). (A1'–D1', A2'–D2') Surface view of the posterior pole of embryos shown in (A1–D1, A2–D2). (A1–D1) Control embryos. (A2–D2) *Gb-Toll1* knockdown (pRNAi) embryos. (A1, A2) Lateral surface views of embryos showing *Gb-zen* expression. (B1, B2) Ventral surface views of embryos showing *Gb-sim* expression. (C1, C2) Ventral surface views of embryos showing *Gb-twi*

*Figure 9 continued on next page*

Figure 9 continued

expression. (D1, D2) Lateral surface views of embryos showing *Gb-wg* expression. Staging according to *Donoughe and Extavour, 2016*; *Sarashina et al., 2005*.

The online version of this article includes the following figure supplement(s) for figure 9:

**Figure supplement 1.** The strong phenotype of *Gb-dl1* knockdown (KD).

**Figure supplement 2.** Expression of *Gb-Toll1* and *Gb-cactus*.

head and *Gb-wg* ocular domains (*Figure 7C2,F2*). These fate shifts are likely to be accompanied by a complete loss of amnion and dorsal ectoderm, cell fates that are characterised by elevated levels of BMP signalling. However, both mesoderm (ventral *Gb-SoxN*, *Gb-zen* and *Gb-twi* expression) and mesectoderm (*Gb-sim*) specifications are almost normal, and the small deviation from controls might be a secondary consequence of the lack of convergent extension (*Figure 7A2,B2,D2*, *Figure 7—figure supplement 2*).

Localised ventrolateral Gb-*SoxN* and *Gb-wg* expression even indicates that the patterning of the ventral region of the neuroectoderm is unaffected after loss of BMP (*Figure 7C2',G2*). This phenotype is similar to *Drosophila* where the loss of BMP does not affect the mesoderm and the ventral-most portion of the neuroectoderm (*Arora and Nusslein-Volhard, 1992*; *Crocker and Erives, 2013*; *Mizutani et al., 2006*; *Ray et al., 1991*; *von Ohlen and Doe, 2000*). In contrast, even a partial reduction of BMP signalling in *Nasonia* and *Oncopeltus* leads to a significant expansion of the mesoderm (*Özüak et al., 2014a*; *Özüak et al., 2014b*; *Sachs et al., 2015*). The possibility exists that our KD is not complete. This could in particular be due to the fact that BMP signalling is required during oogenesis like in *Drosophila* (*Dolezal and Pignoni, 2015*; *Kirilly et al., 2005*; *Upadhyay et al., 2017*), and hence parental RNAi causes sterility. *Gb-dpp2* KD, for which we did not observe sterility, might only cause a partial loss of BMP signalling as the two other BMP ligands (*Gb-dpp1* and *Gb-gbb*) are still functioning. For KD of the single BMP receptor *Gb-tkv*, the few surviving eggs might be produced from ovaries with residual BMP activity. Although we currently cannot rule out these alternative explanations, we regard them as unlikely given the ease with which we have detected the

expansion of the mesoderm by parental RNAi in *Nasonia* and *Oncopeltus*, where reduced BMP signalling also causes sterility (*Özüak et al., 2014b*; *Sachs et al., 2015*).

Second, in *Gryllus* blastoderm embryos, BMP signalling forms a long-range gradient with peak levels at the dorsal and diminishing levels towards the ventral side (*Figure 5*). Similar BMP signalling gradients have been observed in *Drosophila*, *Tribolium* and *Nasonia* (*Dorfman and Shilo, 2001*; *Özüak et al., 2014b*; *van der Zee et al., 2006*) while in *Oncopeltus* early BMP signalling is less graded and has more of a plateau-like distribution (*Sachs et al., 2015*; *Figure 1*). However, while in *Nasonia* Toll is not involved in polarising BMP signalling, in *Drosophila* and *Tribolium* ventral Toll activation triggers the formation of the BMP gradient. In *Drosophila*, BMP gradient formation is bi-phasic. The first Toll-dependent phase results in a broad relatively flat distribution while auto-feedback of BMP signalling in the second phase leads to a narrow dorsal peak of high signalling activity (*Gavin-Smyth et al., 2013*; *Wang and Ferguson, 2005*). How the graded distribution of BMP signalling in *Gryllus* is established is unknown, in particular since one of the major components

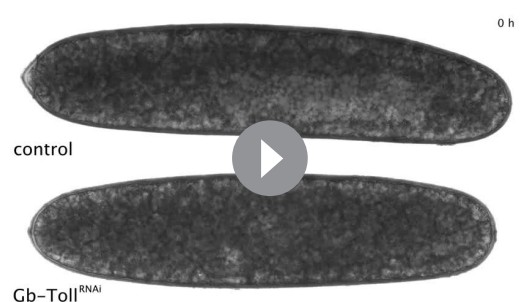

**Video 6.** Complete development of embryos lacking Toll signalling (*Gb-Toll1* knockdown [KD]). Time-lapse imaging of a control and a *Gb-Toll1* RNAi embryo from egg stage 8 onwards. The serosa detaches from the posterior pole in both embryos (starting at 20 hr). While the control embryo undergoes katatrepsis (starting at 53 hr), the serosa of the *Gb-Toll1* RNAi embryo also detaches from the anterior (starting at 72 hr). This leads to a compaction of the yolk within the centre of the egg. This represents the strong phenotype that is observed after the KD of *Gb-Toll1*, *Gb-dl1* and *Gb-spz*. Yolk is taken up into the gut in the control embryo (starting at 110 hr), and the movie ends at egg stage 19. Staging according to *Donoughe and Extavour, 2016*.

https://elifesciences.org/articles/68287#video6

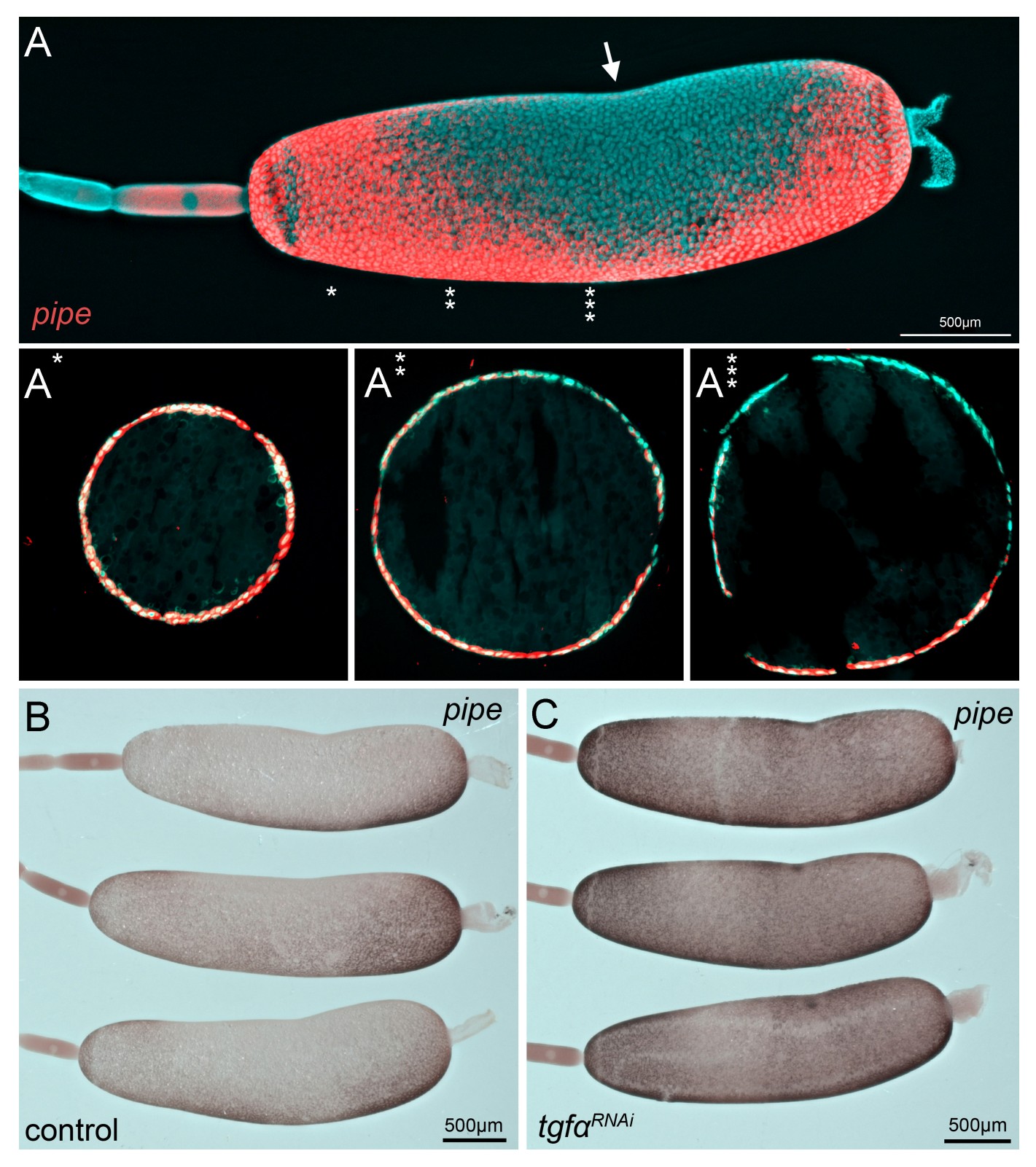

**Figure 10.** *Gb-pipe* expression in oocytes. (**A**) A part of an ovariole of *Gryllus bimaculatus* showing the expression of *Gb-pipe* at the final stages of oogenesis. *Gb-pipe* is expressed in a ventral-to-dorsal gradient within the follicle cells of the late oocyte. In *Gryllus*, the kink within the oocyte (arrow in **A**) is an indicator for the position of the oocyte nucleus. (**A\*–A\*\*\***) Transverse sections of the oocyte at the indicated positions (see \* in **A**). Images are

*Figure 10 continued on next page*

*Figure 10 continued*

false-colour overlays of in situ hybridisation images. **(B, C)** *Gb-pipe* expression in control **(B)** and *Gb-Tgfα* RNAi **(C)** oocytes. In *Gb-Tgfα* RNAi oocytes, *Gb-pipe* is no longer repressed from dorsal follicle cells.

The online version of this article includes the following figure supplement(s) for figure 10:

**Figure supplement 1.** Follicle cell expression of *pipe* and *nudel* in different insects.

involved in this process in *Drosophila* and *Tribolium*, the BMP inhibitor *sog*, seems to be absent from the *Gryllus* genome (see below). However, the phenotypic analysis clearly shows that Toll signalling is required for BMP gradient formation and that this process is not restricted to the late

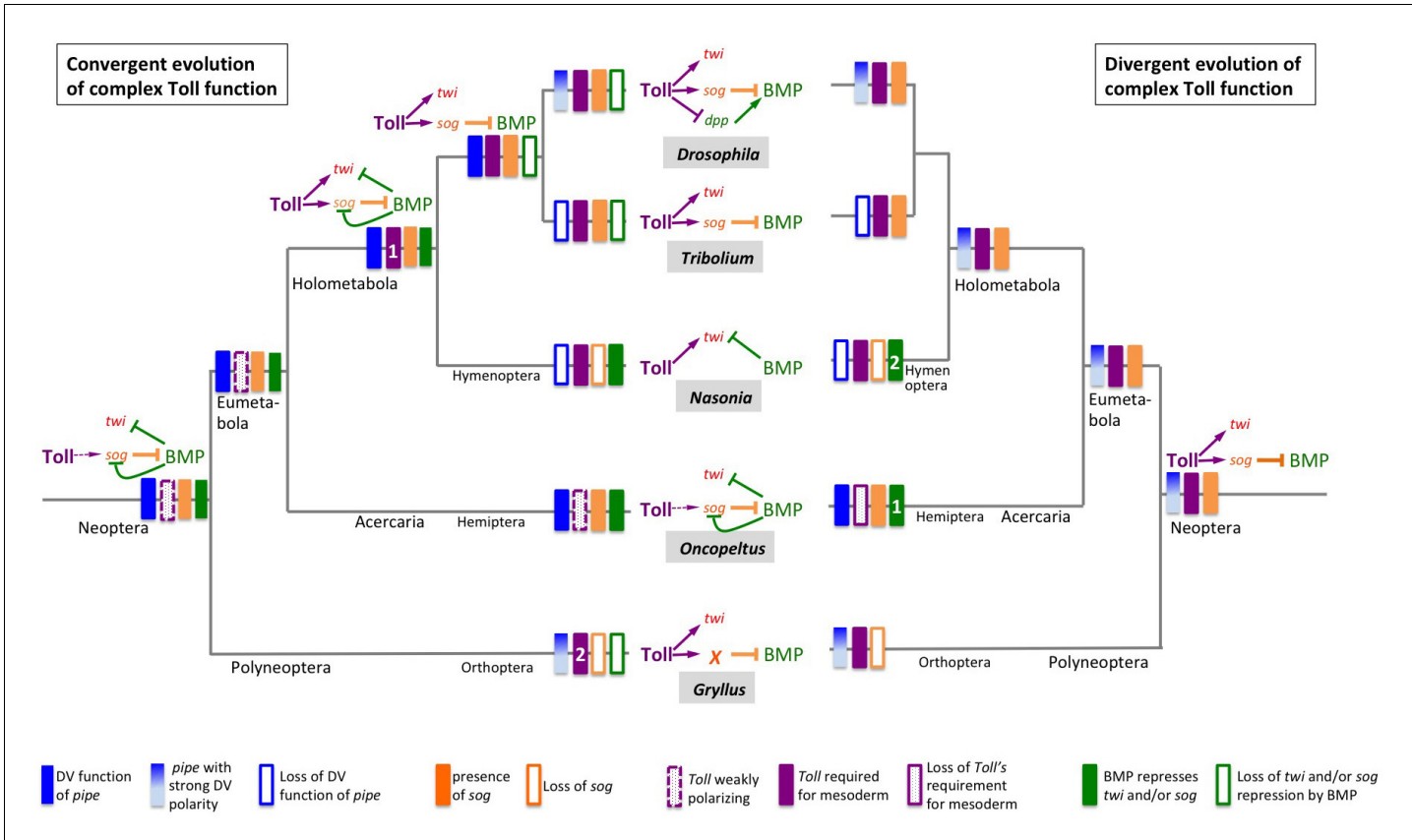

**Figure 11.** Two evolutionary scenarios for the role of Toll signalling in *Gryllus*. **Left:** Toll gained its role for specifying ventral fates including the mesoderm by convergent evolution. Toll signalling in the Neopteran ancestor was weakly polarised by ovarian *pipe* expression and served as polarity cue for self-organising BMP signalling. Toll was not directly involved in mesoderm formation. This mode of dorsoventral (DV) patterning was maintained in the lineage leading to the Hemiptera. At the base of the Holometabola, a strict requirement of Toll for mesoderm formation emerged (1) while BMP remained to be involved in patterning ventral cell fates including the mesoderm. This situation was partially maintained in hymenoptera, like *Nasonia*. In the common ancestor of beetles and flies, the function of BMP signalling became restricted to the dorsal side while Toll patterned the ventral side in a BMP-independent manner. In flies, multiple parallel mechanisms evolved by which Toll restricts BMP signalling to the dorsal side, like the transcriptional repression of *decapentaplegic* (*dpp*), the gene coding the fly BMP2/4 homolog. In addition, ovarian *pipe* expression became spatially refined along the DV axis. In other holometabolan lineages (*Tribolium*, *Nasonia*), ovarian expression of *pipe* was lost and replaced by unknown mechanisms. The polyneopteran lineage leading to *Gryllus* represents another case (2) where Toll gained essential BMP-independent functions in specifying ventral fates including the mesoderm and in which ovarian *pipe* expression became spatially refined along the DV axis. However, in this lineage *sog* was lost and replaced by an unknown gene (or several genes) responsible for Toll-dependent BMP inhibition. **Right:** Toll possessed an essential role in specifying ventral fates already in the Neopteran ancestor. Expression of *pipe* was already strongly polarised along the DV egg chamber axis. This situation was maintained in the lineages leading to *Drosophila* and *Gryllus*. However, within the Gryllidae, *sog* was replaced by an unknown BMP inhibitor. In the linage leading to the Hemiptera, *pipe* expression lost spatial precision and Toll lost both spatial precision and its essential role for ventral cell fate specification. In Hemiptera (1) and Hymenoptera (2), BMP independently gained patterning functions for ventral cell fates. Ovarian expression of *pipe* was lost and replaced by unknown mechanisms in lineages leading to *Nasonia* and *Tribolium*.

condensing germ anlage, but starts at the early differentiated blastoderm stage along a broad region of the AP egg axis (*Figure 5*). This suggests that Toll signalling also occurs globally throughout the *Gryllus* egg similar to *Drosophila* and *Tribolium* (*Figure 1*).

Third, determination of spatial coordinates and specification of mesodermal, mesectodermal and ventral neuroectodermal cell fates depends on Toll signalling in *Gryllus*. This strongly suggests that Toll signalling not only provides the instructive signals for mesoderm specification, but also spatial information, which at least is sufficient to determine the border of the mesoderm (*Figure 1*). Mesectoderm and ventral neuroectoderm specification might then result from secondary interactions, as is the case to some extent in *Drosophila* and *Tribolium* (*Zinzen et al., 2006*). Thus, Toll is likely to act as concentration-dependent morphogen in *Gryllus*. This idea is also supported by our observation of weak KD phenotypes characterised by partial loss of ventral and expanded dorsal tissue regions (*Figure 7—figure supplement 3A–D*). For Toll pathway components, such phenotypes have so far only been observed in *Drosophila* and can be explained by coordinated cell fate shifts due to the decreased slope of a morphogen gradient (*Anderson et al., 1985a*; *Nüsslein-Volhard et al., 1980*; *Roth et al., 1991*).

Fourth, the Toll signalling gradient in *Gryllus* appears to originate in a similar way as in *Drosophila*. So far we have observed two mechanisms that can account for precise long-range Toll signalling gradients. In *Drosophila*, highly accurate spatial information is provided via the eggshell: the ventral half of the inner eggshell (vitelline membrane) carries glycoproteins, which are sulfated by the sulfotransferase Pipe (*Sen et al., 1998*; *Stein and Stevens, 2014*; *Zhang et al., 2009a*; *Figure 1*). These sulfated proteins trigger an extraembryonic protease cascade which activates the ligand of the Toll receptor (*Stein and Stevens, 2014*). In *Tribolium*, Toll signalling seems to receive little spatial information from the eggshell and *Tc-pipe* is not expressed in the ovary (*Figure 1*, *Figure 10—figure supplement 1B*). Toll gradient formation depends on extensive zygotic feedback and has self-regulatory features (*Lynch et al., 2010*; *Nunes da Fonseca et al., 2008*). Expression analyses of Toll pathway components in *Gryllus* do not provide evidence for zygotic feedback mechanisms (*Figure 9—figure supplement 2*). However, in contrast to *Tribolium* and similar to *Drosophila, Gb-pipe* is present in *Gryllus* and shows highly localised expression in the ventral follicular epithelium (*Figures 1* and *10*, *Figure 10—figure supplement 1*).

Due to problems with lethality, we have so far been unable to prove that *Gb-pipe* is involved in DV patterning. Indeed, *Drosophila pipe* also has late functions which compromise larval viability (*Zhang et al., 2009b*). However, *Gb-pipe* expression, together with another follicle cell-specific gene, required for DV patterning in *Drosophila*, the protease Ndl (*Hong and Hashimoto, 1995*), makes an involvement of *Gb-pipe* in providing spatial information for Toll signalling very likely. Moreover, like *Drosophila pipe*, *Gb-pipe* is negatively regulated by EGF signalling, resulting in a clear on-off pattern along the DV egg axis (*Figure 10*; *Sen et al., 1998*). Such a strong negative regulation of *pipe* by EGF signalling is not apparent in *Oncopeltus* where *Of-pipe* distribution seems to be uniform or only weakly asymmetric in the follicular epithelium (*Figure 10—figure supplement 1B*). In *Oncopeltus*, we could show that *Of-pipe,* together with *Of-ndl*, is required for Toll activation (*Chen, 2015*). The lack of obvious asymmetry of *Of-pipe* expression is in line with the model of DV patterning in *Oncopeltus*, where Toll does not act as morphogen, but only as a weak polarity cue (*Sachs et al., 2015*). Likewise, the observation of strongly asymmetric *Gb-pipe* expression is in line with the suggested morphogen function of *Gryllus* Toll signalling.

## Developmental and molecular differences between *Gryllus* and *Drosophila*

The parallels between *Gryllus* and *Drosophila* DV patterning both at the level of the embryo (restriction of BMP signalling to dorsal cell fates, Toll-dependent polarisation of BMP signalling, Toll-dependent ventral cell fate determination) and at the level of the egg chamber (EGF-dependent local *pipe* expression) suggest that among all insects studied so far the regulatory logic of the *Gryllus* DV system is closest to that of *Drosophila* (*Figures 1* and *11*).

This outcome is surprising as the patterning environment of *Gryllus* ovaries and embryos is very different from that of Drosophila. The *Gryllus* egg is about 40 times larger (in volume) than the *Drosophila* egg and develops in an ovary lacking nurse cells (panoistic) (*Sarashina et al., 2005*; *Zeng et al., 2013*). *Gb-pipe* expression is restricted to the ventral follicle cells by repressive EGF signalling that emanates from the dorsally localised oocyte nucleus. However, in contrast to *Drosophila*

the RNA of the EGF ligand is not localised to the oocyte nucleus so that the mechanism of local EGF signalling remains elusive (*Lynch et al., 2010*).

Even more striking are differences during early embryogenesis. *Gryllus* development is about 10 times slower than that of *Drosophila* (gastrulation occurs at about 30 hr instead of 3 hr after egg laying) (*Donoughe and Extavour, 2016*; *Sarashina et al., 2005*). In *Drosophila*, the blastoderm nuclei and later cells hardly move and almost all of them give rise to embryonic structures, with the exception of a narrow stripe of cells straddling the dorsal midline (between 20% and 60% egg length) (*Campos-Ortega and Hartenstein, 1997*). In *Gryllus*, most of the blastoderm cells go through a stage of large-scale movements (condensation) that generate an extended anterior and dorsal region giving rise to serosa and a small embryonic anlage (*Nakamura et al., 2010*). In *Drosophila*, all segments, including their subdivision along the DV axis, are specified almost simultaneously before gastrulation (long germ development) (*Campos-Ortega and Hartenstein, 1997*). In contrast, the embryonic anlage of *Gryllus* gives rise only to head and anterior thoracic segments while the remaining segments form from a posterior segment addition zone after gastrulation (*Miyawaki et al., 2004*).

The long-germ mode of development in *Drosophila* necessitates a high degree of fine-grained spatial information along both the AP and DV body axis. The precision of the Toll signalling gradient, which is reflected in the graded nuclear distribution of the downstream transcription factor NF-κB/Dorsal, seems ideally suited to provide the required spatial accuracy (*Schloop et al., 2020*). However, our results show that the *Gryllus* embryo also makes extensive use of Toll signalling. Correct embryo condensation requires Toll signalling (*Figure 6*), suggesting that a large-scale Toll signalling gradient is present prior to the beginning of condensation (lateral plate formation). It is possible that the cells acquire the accurate DV positional information at this early stage and maintain this information during subsequent migration by epigenetic mechanisms. Alternatively, the Toll signalling gradient might dynamically change like in *Tribolium*, where it becomes progressively more narrow and steeper during embryo condensation (*Chen et al., 2000*). The gradient thus might accommodate its range to the scale of the condensing embryonic anlage being able to continuously provide patterning information. However, as mentioned above, so far we have no data supporting zygotic feedback control of Toll signalling in *Gryllus*.

One of the most striking molecular deviations from *Drosophila* and *Tribolium* DV patterning concerns is the absence of *sog* from *Gryllus*. As Toll is required to set up the BMP gradient in *Gryllus*, one has to postulate that Toll either directly inhibits BMP signalling components, activates (one or more) BMP inhibitors or fulfils both functions simultaneously like in *Drosophila* (*Jaźwińska et al., 1999*; *O'Connor et al., 2006*). Although Sog/Chordin is the most conserved BMP inhibitor for axial patterning in Metazoa, it can work in parallel to or be replaced by other BMP inhibitors or alternative polarising mechanisms. For example in frogs, the KD of three BMP inhibitors (Chordin, Noggin and Follistatin) is required to effectively prevent BMP inhibition (*De Robertis, 2009*). During axial patterning in planarians, Noggin and inhibition of ADMP by BMP signalling play important roles for establishing BMP polarity (*Gaviño and Reddien, 2011*; *Molina et al., 2011*) while in the leech Helobdella the alternate BMP inhibitor Gremlin is involved (*Kuo and Weisblat, 2011*). The *Gryllus* genome contains orthologs of ADMP, Noggin, Follistatin and Gremlin as well as an ortholog of an inhibitory BMP ligand (BMP3) (*Figure 2—figure supplements 1*, *5* and *7*). So far, the KD of these components singly has not produced patterning defects. It is also possible that within the Polyneopterans a new mechanism of BMP inhibition has evolved which we would not be able to discover through a candidate gene approach. Despite the fact that the loss of Sog is only evident for Gryllidae, we think it is likely that Sog function has changed before the gene was actually lost. This is the evolutionary pattern observed in Hymenoptera where *sog* is still present in many lineages; however, its late expression pattern excludes it from contributing to Toll-dependent DV patterning (*Özüak et al., 2014b*; *Wilson et al., 2014*). Thus, it will be interesting to study Sog expression and function in embryos of other polyneopteran species (e.g. cockroaches) to test for changed roles in DV patterning.

## Are the similarities of *Drosophila* and *Gryllus* DV patterning the result of convergent evolution?

Phylogenetically, *Gryllus* belongs to the Polyneoptera, a sister group of the Eumetabola, a clade that includes *Oncopeltus* and all holometabolous insects (*Misof et al., 2014*; *Wipfler et al., 2019*; *Figures 1* and *11*). In *Oncopeltus*, Toll signalling acts only indirectly via BMP and apparently provides

little patterning information (*Sachs et al., 2015*), while in *Gryllus* Toll has direct patterning functions like in holometabolous insects and apparently even acts as a morphogen like in *Tribolium* and *Drosophila*. In principle, this scenario may have two explanations. Toll might have had a DV morphogen function already at the base of the Neoptera (*Figure 11*). The lack of this function in *Oncopeltus* might be a derived state in which Toll's instructive function was replaced by self-organising BMP signalling. Indeed, the most convincing demonstration of self-regulative patterning along the DV axis is the experimentally induced embryonic twinning in the leaf hopper Euscelis which, like *Oncopeltus*, belongs to the Hemiptera (*Sander, 1971*). In almost all other cases where embryo duplications have been observed, it is not clear whether this is due to self-regulatory properties of early DV patterning (*Sander, 1976*). Thus, we cannot exclude that the patterning system of *Oncopeltus* is a novelty of the Hemiptera. However, BMP is also responsible for restricting ventral fates, including the mesoderm, in *Nasonia*. If a *Drosophila*-like DV patterning system emerged early and was maintained in the lineage leading to *Drosophila*, the requirement of BMP signalling for controlling ventral fates would have evolved independently on at least two occasions: once within Hemiptera and once within Hymenoptera (*Figure 11*).

Given the larger phylogenetic context, we regard it, however, as unlikely that an extended requirement of BMP signalling for DV patterning represents a derived state. Toll signalling has not been found to be involved in DV patterning outside the insects while BMP signalling is, like in *Oncopeltus*, essential during axial patterning and has self-regulatory features in many other animals (*Bier and De Robertis, 2015*). Even specific aspects of the *Oncopeltus* GRN like the BMP-dependent inhibition of *Of-sog* expression are widely shared with other metazoan lineages including basal arthropods (*De Robertis and Moriyama, 2016*; *Genikhovich et al., 2015*). Therefore, we would expect an *Oncopeltus*-like GRN to be found at the base of the insects and that the *Nasonia* system represents an intermediate state in which ventral cell fates require Toll for specification and BMP for patterning (*Figures 1* and *11*). This implies that the *Gryllus* GRN represents a case in which Toll has acquired a morphogen function independently from the holometabolous lineages.

Interestingly, recent work on comparative embryology has pointed out that some features of early polyneopteran development related to DV patterning represent derived states (apomorphies) (*Wipfler et al., 2019*). Basally branching wingless (Zygentoma and Archaeognatha) and palaeopteran (dragonflies, damselflies, mayflies) insects form the germ anlage by a simple condensation and proliferation of blastoderm cells near the posterior pole of the egg. This is also observed in many Acercaria (which include the Hemiptera). In most polyneopteran orders, however, germ anlage formation begins with the formation of clearly demarcated lateral regions with higher cell density, which subsequently move to the ventral side (*Mashimo et al., 2014*). Therefore, formation and subsequent fusion of paired blastoderm regions was suggested as an apomorphy of the Polyneoptera, while simple germ anlagen condensation was considered to be part of the ectognathan groundplan (*Wipfler et al., 2019*). The formation of clearly demarcated lateral blastoderm regions requires DV patterning information spanning the entire embryonic circumference. We have shown here that in *Gryllus* this is accomplished by direct and indirect (BMP-mediated) actions of Toll signalling linking this apomorphy to properties of the Toll GRN.

A second peculiarity of polyneopteran development concerns the positioning of the elongating embryo within the egg (*Wipfler et al., 2019*). While in Palaeoptera, Acercaria and even many holometabolous insects like *Tribolium* embryo elongation is linked to its invagination into the yolk, the embryo stays at the egg surface during elongation in Polyneoptera (*Mashimo et al., 2014*). This has implications for the timing of amnion specification. When the embryo sinks into the yolk, amnion specification might be quite late during germ band elongation (*Benton, 2018*). As the polyneopteran embryo stays flat at the egg surface, the cells that migrate over the ventral side of the embryo to form the amniotic cavity acquire characteristics of amniotic cells very early (*Dearden et al., 2000*; *Roonwal, 1936*). Accordingly, spatial patterning, giving rise to the distinction between amnion and dorsal ectoderm, has to occur early during development. This requirement is in line with the steep pMad gradient we have observed in early *Gryllus* embryos which ultimately depends on an opposing Toll signalling gradient. Thus, also this polyneopteran apomorphy can be related to particular features of the DV GRN in *Gryllus*.

Considering the minimal role Toll plays in *Oncopeltus* and the developmental apomorphies of Polyneoptera, which are in line with a GRN providing fine-grained DV patterning information, it is most parsimonious to conclude that the employment of Toll signalling in *Gryllus* represents a case of

independent evolution. This idea is also supported by the unusual loss of sog/chordin suggesting a different spectrum of target genes being regulated by *Gb-Toll* compared to *Drosophila*. Taken together, the *Gryllus* DV system provides a unique opportunity to study parallel evolution in one of the best-understood model regulatory networks in animal development. Future work will make use of established transgenesis in *Gryllus* to perform live imaging of Toll and BMP gradient formation, as well as genomic approaches combined with RNAi to identify Toll and BMP target genes in an unbiased way, adding further detail to our knowledge of how Toll signalling was recruited as a DV morphogen.

## Materials and methods

### G. bimaculatus husbandry

*G. bimaculatus* wildtype strain and pXLBGact Histone2B:eGFP (*Nakamura et al., 2010*) were kept at 27°C with 10 hr light and 14 hr dark cycle and were fed with artificial fish and cricket food and dried mealworms. Eggs were collected in moist floral foam (big-mosy, Mosy, Thedinghausen, Germany).

### Gene cloning

PCR amplification was carried out using the Advantage GC 2 PCR Kit (Clontech Catalog #639119), and cloning was carried out using standard procedures (primer sequences are listed below).

### RNA and read sources

For sequencing the embryonic transcriptome, we extracted total RNA of mixed stage embryos of the wt (until egg stage 12 [staging according to *Donoughe and Extavour, 2016*], combined into one sample, known as Gb1) and egg stages 1–3 (combined into one sample, known as Gb2) of the pXLBGact Histone2B:eGFP strain (*Nakamura et al., 2010*). An ovary was dissected out of an adult female, and total RNA was extracted as described below.

For total RNA extraction, 50–100 mg of embryos or ovaries were homogenised in 1 ml of TRIZOL. After centrifugation for 15 min at 4°C and 12,000 g, the supernatant was transferred to a fresh tube and 200 μl chloroform/1 ml of TRIZOL was added. Vortexing for 15 s and centrifugation for 15 min at 4°C and 12,000 g led to a phase separation. The total RNA of the upper aqueous phase was cleaned using the Zymoclean Quick RNA MicroPrep Kit according to the manufacturer's protocol. Library preparation (including poly A selection) and stranded sequencing was carried out at the Cologne Center for Genomics (HiSeq2000 for the embryonic and HiSeq4000 for the ovarian transcriptome).

We also downloaded previous sequencing reads from transcriptomic studies of *G. bimaculatus* (*Fisher et al., 2018*) from the sequence read archive (SRA). The previously sequenced reads and those from our sample were subjected to adaptor and quality threshold trimming using Trimmomatic 0.33 (*Bolger et al., 2014*), the appropriate adaptor sequences and the following quality cutoffs: LEADING:3 TRAILING:3 SLIDINGWINDOW:4:15 MINLEN:30. FastQC (*Andrews, 2010*) was used to assess read quality both before and after trimming. Raw reads from our sequencing are available from the NCBI SRA under accession PRJNA492804.

### Assembly and annotation

The trimmed read sets from both our data and previous sources were used to make a combined assembly using Trinity 2.6.6 (*Grabherr et al., 2011*). Reads were normalised with a maximum coverage of 50. In the process of assembly checks, we noted a small amount of *Parasteatoda tepidariorum* contamination of our transcriptome assembly, likely the result of adaptor mis-assignment (*P. tepidariorum* was sequenced on the same lane as our ovary sample). This has been removed by comparison to the *P. tepidariorum* genome (*Schwager et al., 2017*) and transcriptome (*Posnien et al., 2014*) resources using megablast and removal of hits, which were empirically obvious, with an *E* value cutoff of 0. The cleaned transcriptome was used for all further analysis. Metrics regarding the resulting assembly were gathered using the TrinityStats.pl script.

Diamond (*Buchfink et al., 2015*) was used to perform BLASTx searches for initial annotation, against a locally downloaded version of the NCBI nr database. An *E* value cutoff of $10^{-6}$ was applied, and the `—max-target-seqs 1 —outfmt` six qseqid stitle evalue `—more-`

`sensitive settings`. BUSCO v1.1b1 (*Simão et al., 2015*) was used to assess our transcriptomic assembly content, using the metazoan and eukaryote datasets. Expression levels for each sample were quantified using RSEM (*Li and Dewey, 2011*) and Bowtie2 (*Langmead and Salzberg, 2012*) as packaged in the Trinity module (align_and_estimate_abundance.pl script,—est_method RSEM—aln_-method bowtie2) to compare individual RNAseq samples with the combined assembly.

## BMP and Toll gene pathway identification

As well as identifying genes automatically via our initial annotation, we manually identified target genes using tBLASTn (*Altschul et al., 1990*) searches. We used gene sequences of known homology downloaded from the NCBI nr database (particularly those of *T. castaneum; Van der Zee et al., 2008*) and from NasoniaBase (*Munoz-Torres et al., 2011*; Nvit_2.1 where possible), as well as those identified in *Kenny et al., 2014* as queries against standalone databases for both *G. bimaculatus* and *O. fasciatus* (GCA_000696205.2, Ofas_2.0) using BLAST 2.2.29+ on a local server. These putatively identified genes were reciprocally BLASTed against the online NCBI nr database using BLASTx to confirm their identity. Novel sequences thus putatively identified were then aligned to sequences of known homology using MAFFT v7 (*Katoh et al., 2019*). The resulting alignments were then subjected to Bayesian analysis in MrBayes v3.2.6 x64 (*Ronquist and Huelsenbeck, 2003*). The model jumping setting was used, selecting substitution models proportionally to posterior probability. Markov chain Monte Carlo searches were run for 1,000,000 generations with sampling every 100 generations, or for as many generations as required for the average standard deviation of split frequencies to be less than 0.01. The first 25% of generations were discarded as burn-in. Trees were then visualised in FigTree (*Rambaut, 2012*) for annotation and display.

## HMM-based identification of Sog fragments in Orthoptera

To detect *sog* genes in sequence sets lacking annotation, we first built a HMM of Sog from annotated insect Sog proteins. We then used the Sog HMM (951 AA, 80 sequences from 77 species) to identify Sog, or Sog fragments, in translated transcriptomic sequences from additional insect species, including orthopterans, and annotated their matches to the Sog HMM (*Supplementary file 1*; matches correspond to blue boxes in *Figure 2*).

In detail, we first fetched known insect Sog protein sequences from NCBI using the NCBI reference sequence of *Drosophila melanogaster sog* (accession: NP_476736; 1038 AA) as query in Blastp searches within the taxonomic group 'true insects' (taxid:50557). From the resulting hits, we collected a set of 80 sequences from 77 distinct species comprising the orders Diptera, Hymenoptera, Thysanoptera, Coleoptera, Lepidoptera, Hemiptera and Blattodea. We aligned the sequences using the MAFFT v7.304b 'einsi' algorithm with default parameters (*Katoh and Standley, 2013*) and removed indels and unalignable regions manually. We then produced a 951-AA-long HMM from the modified alignment using the command 'hmmbuild' as provided by the HMMer software suite version 3.1b2 (*Eddy, 2011*).

To discover potential Sog candidates in the order Orthoptera, we downloaded from NCBI at https://www.ncbi.nlm.nih.gov/Traces/wgs/?page=1&view=tsa 28 publicly available transcriptome shotgun assemblies (TSAs) from 22 orthopteran species by restricting the search to the taxonomic group 'Orthoptera' and to project type 'TSA' (*Supplementary file 1*). These transcriptomes included 11 orthopteran TSAs present in the dataset of *Wipfler et al., 2019* and 5 orthopteran transcriptomes published in the dataset of *Misof et al., 2014*. We translated the 28 TSAs into all possible reading frames using the Emboss tool 'getorf' with parameter '-minsize 84' (*Rice et al., 2000*). Finally, we scanned the resulting transcriptome ORFs with the Sog HMM using the command 'hmmsearch' (*Eddy, 2011*) and verified the similarity of hit ORFs to known Sog proteins by reciprocal best hit BLAST searches at NCBI (column 'R' in File S1 'allbusco_allstat_orthoptera4.xlsx').

To evaluate the reliability of Sog presence/absence in a species, we assessed the completeness of all collected transcriptomes using the BUSCO pipeline (*Simão et al., 2015*) with the OrthoDB dataset 'insecta_odb9' (1658 orthogroups from 42 insect species; https://busco-archive.ezlab.org/v3/datasets/insecta_odb9.tar.gz) as reference gene set. According to this reference, the average transcriptome completeness of the entire dataset (29 transcriptomes) was 80.14%. Within Gryllidae (six transcriptomes), average completeness was >90% despite the poor transcriptome of *Gryllus firmus* (37.9% missing BUSCOs).

We annotated Sog in an analogous way in the transcriptome GDCR01 from *B. germanica* (out-group: Blattodea). As two additional outgroup sequences, we scanned the Sog proteins of the may-fly *Ephemera danica* (NCBI accession: KAF4519369) and of the springtail *Folsomia candida* (uniprot ID: A0A226CYM6) with the Sog HMM and annotated regions of similarity. Finally, we mapped the Sog fragments identified in orthopteran and outgroup sequences onto a schematic phylogenetic tree using Affinity Designer v1.73.

## Parental RNAi experiments

For parental RNAi experiments, adult females of the wildtype and the pXLBGact Histone2B:eGFP (*Nakamura et al., 2010*) strain were injected with dsRNA and mated to wildtype males. *Gb-Toll1* gene function was tested by independently injecting three non-overlapping dsRNA probes (5′, mid and 3′ fragments). For *Gb-dpp1*, *Gb-dpp2*, *Gb-tkv* and *Gb-dorsal* RNAi, a single fragment per gene paralog was used to analyse the gene function. For all RNAi experiments, 10 µg of dsRNA per adult female was injected. The maximal volume of the injected dsRNA solution was 5 µl. Water injections (5 µl) served as a control.

To observe the influence of *Gb-tgfα* on oocyte polarisation, we injected egg-laying adult female crickets with dsRNA (10 µg/female) and dissected and fixed the ovaries 20 hr after injection.

## In situ hybridisation, pMad antibody staining and embryo fixation

In situ hybridisation (ISH) was performed as described in *Kainz, 2009*.

To analyse BMP pathway activity in control and *Gb-Toll* KD embryos, pMad antibody staining was performed as described in *Pechmann et al., 2017*.

For the pMad antibody staining, early cricket embryos (until egg stage 5) were dechorionated using 50% bleach (DanKlorix). Subsequently, embryos were washed several times with water until the embryos floated to the top of the water surface. Water was removed and embryos were fixed for 20 min in heptane/PBS containing 5% formaldehyde. The fixative was removed completely and the vitellin membrane was carefully removed in PBST using fine forceps (Dumont #5). Devitellinised embryos were transferred to PBST containing 5% formaldehyde and were fixed for overnight.

In situ hybridisation on blastoderm and early germ band embryos (until egg stage 5) was per-formed on heat-fixed embryos. For this, we collected 20–30 eggs in a 1.5 ml Eppendorf-tube. The eggs were washed two times with distilled water. Embryos were heat fixed in 100 µl distilled water at 99°C for 90 s and cooled down on ice for 2–5 min. To crack the eggshell, embryos were washed two times with 100% methanol. After the methanol shock, embryos were transferred to a glass dish and the vitelline membrane was removed using sharp needles and forceps. Peeled embryos were transferred to a fixative (5% formaldehyde in 0.01% PBST) and were fixed for several hours to over-night. Embryos were washed with 0.01% PBST and were gradually transferred to 100% MeOH for long-time storage at −20°C.

For in situ hybridisation on embryos of egg stages > 6, living embryos were dissected out of the vitelline membrane using sharp needles and forceps in PBST. Yolk was removed and embryos were fixed in PBST containing 5% formaldehyde overnight.

## Cuticle preparations and fuchsin staining

For cuticle preparation, *Gryllus* eggs were placed into PBST and the vitelline membrane was carefully opened and removed using forceps. Subsequently, larvae were placed on a microscope slide, cov-ered with Hoyer's mounting medium and a cover slip and were incubated at 60°C for overnight.

Fuchsin staining was performed as described in *Miyawaki et al., 2004*.

## Ovary dissection and fixation and durcupan sectioning

Adult *Gryllus* females were paralysed using $CO_2$. Ovaries were dissected quickly in ice-cold PBS with fine forceps and were fixed for overnight in a 5% formaldehyde/PBST solution (*Lynch et al., 2010*). Durcupan sections on *Gb-pipe*-stained oocytes were performed as described in *Pechmann et al., 2017* with minor modifications.

## Imaging and image analysis

All bright field and fluorescent images were recorded using an AxioZoom.V16 microscope (Zeiss), equipped with a movable stage. Phase-contrast images of cuticle preparations were recorded using an Axioplan2 (Zeiss).

For live imaging, eggs were carefully removed from the floral foam and washed with tap water. Subsequently, eggs were placed on 1.5% agarose and covered with Voltalef H10S oil. Live imaging was performed at 25–27°C.

Projections of image stacks were carried out using Helicon Focus (HeliconSoft). Movies were created by using Fiji (*Schindelin et al., 2012*). Images have been adjusted for brightness and contrast using Adobe Photoshop CS5.1.

False-colour overlays of in situ hybridisation images were generated as described in *Pechmann et al., 2017*.

## Primer

Gb-Toll1-5′-Fw: 5′-CAA GAT GGT TGC AAT CAG C-3′
Gb-Toll1-5′-Rev: 5′-CAT GAA TGT GTC AGG TGA TAT GTA G-3′
Gb-Toll1-mid-Fw: 5′-GCA ACA ATA AAA TAG CTT CCA TTC ATG-3′
Gb-Toll1-mid-Rev: 5′-GAA TGC AGC GAG TTG TTG TCT AG-3′
Gb-Toll1-3′-Fw: 5′-CGC TAG TTC AAT TAC AGC TTT ATT AG-3′
Gb-Toll1-3′-Rev: 5′-GAG TTA TCA CTT ATT CAC AAT CAC TC-3′
Gb-dorsal1-Fw: 5′-GCA GCA GTT CAG CAG ACT AC-3′
Gb-dorsal1-Rev: 5′-CTG AAT GAA ATT GCA ACC TG-3′
Gb-dorsal2-Fw: 5′-CAT ATC AGT GAT GTC ATT GAA GTG-3′
Gb-dorsal2-Rev: 5′-GTA CGA AGT GAA ATT GCG AC-3′
Gb-dorsal3-Fw: 5′-GTA GTA TCA ACT CCC AAA CCA ATT G-3′
Gb-dorsal3-Rev: 5′-GTA AAG AAG ATC GAA CAG GTG CC-3′
Gb-twist-Fw: 5′-CAA GCC GCA CTA CGC TTA C-3′
Gb-twist-Rev: 5′-CCT GGT ACA GGA AGT CGA TGT AG-3′
Gb-single-minded-Fw: 5′-GTC TAA TTT TGT CGG TCC AGG-3′
Gb-single-minded-Rev: 5′-CAC GTA GTT CAC GCT GAC G-3′
Gb-pipe-F1: 5′-GGT CGG CCA GTG CCC TGC-3′
Gb-pipe-R1: 5′-GAG TTG CTT GTG GAG TCG TTG G-3′
Gb-pipe-NestedPCR-F2: 5′-CGG CTA TGG ACA GTA TGT TCC G-3′
Gb-pipe-NestedPCR-R2: 5′-CAG AAC TGG TAG AAT TCA ACT TCA C-3′
Gb-tgfα-F1: 5′-GAC GTG TTA GTG GCG ACG CTG-3′
Gb-tgfα-R1: 5′-CAC TGC TGC GAG TGT CGA TGC-3′
Gb-tgfα-NestedPCR-F2: 5′-CGG ACC GCG ACG TGT CTG-3′
Gb-tgfα-NestedPCR-R2: 5′-GTG CTT GAA GGA GCT TCT CCT AC-3′
Gb-dpp1-F: 5′- GAC GTC CTC GAG GGA GAG-3′
Gb-dpp1-R: 5′-GAC AAC CTT GTC CGT CTC G-3′
Gb-dpp2-F: 5′-TTA TGT ACG CGT GGA TGA CG-3′
Gb-dpp2-R: 5′-CGT CTG CTT TCA AGA TCA GG-3′
Gb-tkv-F: 5′-GAC ATC GAC CTT TGC AAC AGA AAC-3′
Gb-tkv-R: 5′-GCA CTG CCA GTC CAA AAT CTG C-3′
Gb-soxN-F: 5′-CCA GCT CCA AGA ACC AGA AC-3′
Gb-soxN-R: 5′-GCT ATG CGA CGA CAT CTG CG-3′

## Acknowledgements

We thank S Noji and T Mito for providing *Gryllus* strains and for information about genomic sequences. O Özüak, T Buchta, J Lynch and Y-T Chen were supported by DFG CRC680. M Pechmann was supported by UoC Postdoc Grant. We thank Kristen Panfilio for discussions and critical reading of the manuscript.

## Additional information

### Funding

| Funder | Grant reference number | Author |
|---|---|---|
| University of Cologne | Postdoc grant | Matthias Pechmann |
| Deutsche Forschungsgemeinschaft | CRC 680 | Yen-Ta Chen<br>Thomas Buchta<br>Orhan Özüak<br>Jeremy Lynch |

The funders had no role in study design, data collection and interpretation, or the decision to submit the work for publication.

### Author contributions

Matthias Pechmann, Conceptualization, Data curation, Formal analysis, Supervision, Funding acquisition, Validation, Investigation, Visualization, Methodology, Writing - original draft, Project administration, Writing - review and editing; Nathan James Kenny, Data curation, Formal analysis, Investigation, Writing - original draft, Writing - review and editing; Laura Pott, Yen-Ta Chen, Thomas Buchta, Orhan Özüak, Investigation; Peter Heger, Formal analysis, Investigation, Writing - original draft; Jeremy Lynch, Investigation, Writing - original draft, Writing - review and editing; Siegfried Roth, Conceptualization, Formal analysis, Supervision, Funding acquisition, Writing - original draft, Project administration, Writing - review and editing

### Author ORCIDs

Matthias Pechmann (iD) https://orcid.org/0000-0002-0043-906X
Nathan James Kenny (iD) https://orcid.org/0000-0003-4816-4103
Laura Pott (iD) http://orcid.org/0000-0002-3314-6239
Peter Heger (iD) http://orcid.org/0000-0003-2583-2981
Jeremy Lynch (iD) https://orcid.org/0000-0001-7625-657X
Siegfried Roth (iD) https://orcid.org/0000-0001-5772-3558

### Decision letter and Author response
Decision letter https://doi.org/10.7554/eLife.68287.sa1
Author response https://doi.org/10.7554/eLife.68287.sa2

## Additional files

### Supplementary files
• Supplementary file 1. Data from hidden Markov model search for short gastrulation homologs in Orthoptera.

• Supplementary file 2. Results from transcriptome annotation based on an automated BLAST approach.

• Supplementary file 3. Sequences of genes coding for BMP and Toll pathway components and their alignments used for phylogenetic reconstruction.

• Supplementary file 4. Recovery of the TGFβ/BMP pathway of *G. bimaculatus*, summary. Where multiple isoforms are present, this is noted in the appropriate column.

• Supplementary file 5. Recovery of Toll pathway components of *G. bimaculatus*.

• Transparent reporting form

### Data availability
Raw reads from our sequencing are available from the NCBI SRA under accession PRJNA492804 The Gryllus transcriptome is available from https://doi.org/10.6084/m9.figshare.14211062.

The following datasets were generated:

| Author(s) | Year | Dataset title | Dataset URL | Database and Identifier |
|---|---|---|---|---|
| Pechmann M, Kenny NJ, Roth S | 2020 | Gryllus bimaculatus raw RNAseq data | https://www.ncbi.nlm.nih.gov/sra/PRJNA492804 | NCBI Sequence Read Archive, PRJNA492804 |
| Pechmann M, Kenny NJ, Roth S | 2021 | Gryllus bimaculatus transcriptome Assembly | https://doi.org/10.6084/m9.figshare.14211062 | figshare, 10.6084/m9.figshare.14211062 |

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
