## [Decision Letter]

[Editors' note: this paper was reviewed by Review Commons.]

**Acceptance summary:**

This is a very fine paper, very well controlled, carefully argued, well written and the findings are of general significance to insect developmental biology, evo-devo and the Dpp/Toll signalling field. The similarity in dorsal/ventral patterning between crickets and *Drosophila* is striking and much greater than expected given differences in patterning between *Drosophila* and more closely related insect species. The authors propose this similarity is due to convergence rather than conservation, which will be very interesting to resolve in future work.

---

## [Author Response]

Reviewer #1 (Evidence, reproducibility and clarity (Required)):Summary:– The authors have carried out an extensive survey of dorso-ventral axis determination in the cricket *Gryllus bimaculatus*. They did this through analysing and knocking down key components of the two main pathways involved in D/V patterning, the toll pathway and BMP signalling. This analysis was placed in a comparative context, looking at published data on four other insect species, with the aim of contributing to our understanding of the evolution of D/V patterning.– The authors find significant similarities between D/V patterning in *Gryllus* and in *Drosophila* – These similarities are both in the relative contributions of toll and BMP to D/V polarization and in the early ovarian activation of the toll pathway. Despite these similarities, a closer analyses of the molecular interactions uncovers some significant differences, most notably, the absence of several key modulators of BMP activity. These results lead the authors to conclude that the similarities in D/V patterning between *Gryllus* and *Drosophila* are due to convergence and not due to the conservation in *Drosophila* of an ancestral patterning mechanism that has been lost in almost all other lineages studied.Major comments:– All in all this is an excellent paper. There is a huge amount of data in here, and everything is done very meticulously and carefully. There is a good balance between mostly descriptive work (gene expression patterns, cell movements in WT embryos) and experimental work. I could find no obvious flaws with any of the results or methods, and I think the authors have made a convincing case to support their conclusions, without being too dogmatic.– I don't see a need for any additional experiments beyond what the authors have done. They have covered all relevant aspects of D/V patterning, and make a convincing case with the data they have.Reviewer #1 Significance (Required)– The manuscript represents an admirable amount of work. One can say that in a single paper, the authors have provided nearly as much information about *Gryllus* D/V patterning as is available for other "second-order" insect model species such as *Oncopeltus* or *Nasonia*. A such, it provides an additional major phylogenetic anchor point for understanding the evolution of early patterning.– In terms of significance to advancing our knowledge, the data in the manuscript is, as stated above, an anchor point. It does not on its own provide any major novel insight, but fits into an ever-expanding body of comparative knowledge, whose importance is greater than the sum of its parts. Perhaps the most interesting conclusion, is indeed the one the authors have chosen as the selling-point of their paper, the fact that there is functional convergence in certain aspects of D/V patterning between two widely diverged insect species, with very different oogenesis and early development. This is again, not a major advance on its own, but an important additional piece of the comparative picture of early insect development.– This paper will be of significant interest to the research community of comparative insect development (the community to which this reviewer belongs). It will also be of interest to those interested in examples of convergence at the functional and molecular level, to those interested in the evolution of gene families and to those interested specifically in the signalling pathways discussed (even in a non-comparative context).Reviewer #2 (Evidence, reproducibility and clarity (Required)):In this paper Pechmann and colleagues investigate the molecular mechanisms of dorso-ventral patterning in *Gryllus bimaculatus*. As a basis for their study they carry out thorough RNAseq analyses of various embryonic stages. *Gryllus* is a member of the hemimetabolous insects and therefore of interest for comparison with holometabolous insects such as Drosophila, *Tribolium* and *Nasonia*. Previous work has shown that there are significant differences in the use of Toll and Sog in establishing the dorso-ventral gradient of BMP signaling among *Drosophila* and *Nasonia*. Pechmann et al. find that in *Gryllus* Toll has a similar role as in *Drosophila* and is regulated via Pipe, so far only found in *Drosophila*. Furthermore, they show by RNAi knockdown studies that loss of BMP signaling has little impact on the differentiation of mesoderm in *Gryllus*, like in *Drosophila*, hence, BMP signaling has largely a role in dorsal fates. Ventral fates are under direct control of the Toll gradient. Surprisingly, they also find that the key antagonist of BMP signaling and shuttle for BMPs, Sog, has been lost in Ensifera, the lineage leading to *Gryllus*.This is a thorough and detailed study involving a series of functional experiments, which highlights the flexibility and evolvability of GRN of the dorso-ventral body axis formation in insects. The major finding that *Gryllus* is more similar to Drosophila than is *Nasonia* and *Tribolium* is interesting and even somewhat unexpected, since *Drosophila* is often regarded as the derived odd ball. The authors discuss two obvious explanations: the situation found in *Gryllus* and *Drosophila* reflects the ancestral condition, or, alternatively, it is the result of convergent evolution. They tend to favor the latter hypothesis. This study is an important advancement to our understanding, as it shows the constraints and the evolvability of a key patterning system to establish a body axis.Even though the authors show nicely that Toll signaling is required to establish the BMP signaling gradient, the loss of Sog in *Gryllus* leaves the question unanswered how the long range BMP gradient and its shape is established. In *Drosophila* and vertebrates, Sog/Chordin acts both as an antagonist close to its source and as a shuttling factor, promoting BMP signaling at a distance, which is crucially important for the long range and the shape of the BMP signaling gradient. It would be desirable to test the function of other potential BMP antagonists (follistatin, gremlin, noggin) or competing BMPs (BMP3, ADAMP) in this context.As a minor suggestion, I would recommend to summarize the findings in a synthetic picture depicting the evolutionary scenarios of the two hypotheses.Reviewer #2 (Significance (Required)):This study is an important advancement to our understanding, as it shows the constraints and the evolvability of a key patterning system to establish a body axis.

As the reviewer suggested we added a schematic representation (Figure 11) depicting the two scenarios, which explain the evolution of DV patterning.

Reviewer #3 (Evidence, reproducibility and clarity (Required)):SummaryThis manuscript continues a series of beautiful papers from Roth, Pechman, Lynch and colleagues analysing D/V patterning in a range of insects. The work started with *Drosophila* and has extended to other holometabolous and now hemimetabolous insect species.This paper is in many ways one of the most remarkable of the series, for it shows that the mechanisms of D/V patterning in the cricket *Gryllus* are, in several striking respects, very similar to those known from *Drosophila* – much more so than in some of the other insects studied to date, even though *Gryllus* is phylogenetically the most distant from *Drosophila*.Specifically, the authors present compelling data to show that the roles of Toll and dpp, as inferred from their knockdown phenotypes, are remarkably similar in *Gryllus* and *Drosophila*. This is very different from the consequences of toll and dpp knockdown in the hemipteran *Oncopeltus*, a species which almost certainly shares a more recent common ancestor with *Drosophila*.The discussion, after summarising the results, addresses the interpretation of this surprising observation. The authors favour the hypothesis that the similarity between *Drosophila* and *Gryllus* is the result of convergence in the roles and regulation of Toll and dpp signalling, rather than an ancestral trait that has been lost to a greater or lesser extent in *Oncopeltus*, and in the two other insects previously studied. The argument for this interpretation is carefully made, on the basis of a thorough knowledge of the comparative embryological literature (including highly relevant recent work).Major commentsThe work depends on an analysis of candidate genes, not de novo functional searches. However, it builds on the well-established understanding of the relevant genetic machinery in *Drosophila*, and on extensive knowledge of the genome and transcriptome of *Gryllus*, a dataset that has been substantially extended by new work reported in this paper, on ovary and embryonic transcriptomes. These data are sufficiently complete to give confidence that all orthologues of most of the known candidate genes have been identified, and to highlight the apparent absence from the *Gryllus* genome of any sog/chordin orthologue – a key dpp inhibitor widely involved in D/v patterning.The embryology is beautifully described. The early stages of these very yolky eggs are not easy to handle, but the stainings reported here are almost all of high quality, as are the videos of live development using a nuclear GFP marked line.The gene knockdowns appear to have been carried out carefully with due regard for the potential biases caused by sterility following parental RNAi. Phenotypes have been documented effectively by the expression of marker genes in fixed embryos, and by live imaging of development in knockdown embryos. Tables in the supplementary data show that sufficient numbers have been obtained. The work is carefully interpreted, and where inferences are less than certain, they are carefully phrased.I find the results convincing, and therefore accept the conclusion of fundamental similarity between the roles of Toll and dpp in *Drosophila* and *Gryllus*.Time will tell whether or not the authors favoured interpretation of these data as convergent is correct, but I certainly believe that the argument as here presented in the discussion is appropriate for publication in its current form. The abstract is, appropriately, more non-committal than the discussion itself on the interpretation of these results.The paper is well written.Reviewer #3 (Significance (Required)):The gene networks mediating patterning of the D/V body axis are related across the whole range of animals, with in particular the involvement of TGFb/dpp signalling being almost universal in this process. However, there are a great many variations on this theme. Even within the insects, the mechanisms that have been described for establishing localised TGFb and Toll signalling span the range from self-organisation to effective maternal prelocalisation. This has made the GRN underlying D/V patterning a key model for studies of the evolution of gene regulatory networks.This paper adds an interesting and important twist to the story. It is certainly not the result that any of us would have expected, based on prior published work from *Oncopeltus*.If indeed it does turn out to be a case of convergence, a more detailed mechanistic analysis of that convergence will provide considerable insight into the reproducibility of evolution.Other published work: There is no comparable work on D/V patterning in any other polyneopteran insect, to my knowledge.Audience: Insect developmental biologists, evolutionary developmental biologists and others interested in the evolution of gene regulatory networks.My expertise: Arthropod embryology, axial patterning, evolutionary developmental biology.